# Imagine the Unseen World: A Benchmark for Systematic Generalization in Visual World Models

**Yeongbin Kim**[*]
KAIST

**Gautam Singh**[*]
Rutgers University

**Junyeong Park**
KAIST

**Caglar Gulcehre**[†]
EPFL & Google DeepMind

**Sungjin Ahn**[‡]
KAIST

## Abstract

Systematic compositionality, or the ability to adapt to novel situations by creating a mental model of the world using reusable pieces of knowledge, remains a significant challenge in machine learning. While there has been considerable progress in the language domain, efforts towards systematic visual imagination, or envisioning the dynamical implications of a visual observation, are in their infancy. We introduce the *Systematic Visual Imagination Benchmark* (SVIB), the first benchmark designed to address this problem head-on. SVIB offers a novel framework for a minimal world modeling problem, where models are evaluated based on their ability to generate one-step image-to-image transformations under a latent world dynamics. The framework provides benefits such as the possibility to jointly optimize for systematic perception and imagination, a range of difficulty levels, and the ability to control the fraction of possible factor combinations used during training. We provide a comprehensive evaluation of various baseline models on SVIB, offering insight into the current state-of-the-art in systematic visual imagination. We hope that this benchmark will help advance visual systematic compositionality. `https://systematic-visual-imagination.github.io`[4]

## 1 Introduction

Constructing a mental model of the world, known as a world model, in a composable way is a crucial aspect of human intelligence [40, 12, 72]. This ability enables humans to adapt to novel situations and problems in a zero-shot manner by envisioning the possible future [79, 7]. Studies in neuroscience and cognitive science suggest that the key to this ability is the process of acquiring abstract, conceptual, and reusable pieces of knowledge from past experiences and applying them in new configurations to comprehend a novel situation [30, 15, 76, 7]. For instance, a person who understands the implications of a scene involving a "big dog" and a "small cat", e.g., how they may interact and what the subsequent scene may look like, can also reasonably understand the implications of an unfamiliar scenario involving a "big cat" and a "small dog". While this capability, termed systematic compositionality, is fundamental to human intelligence, how neural networks can acquire such an ability remains one of the grand challenges in machine learning [63, 6, 36].

The notion of systematic compositionality originates from the fields of philosophy of language and linguistics [95, 30]. Language inherently provides a compositional representation using token

---

[*]Equal Contribution.

[†]This work was partly done while C.G. was at Google DeepMind. C.G. is currently affiliated with EPFL.

[‡]Correspondence to `sungjin.ahn@kaist.ac.kr`.

[4]This is the official project page. It provides links to the benchmark and the code.

37th Conference on Neural Information Processing Systems (NeurIPS 2023) Track on Datasets and Benchmarks.

structures, such as words or characters, which simplifies addressing compositional systematicity. As a result, the AI community has also made efforts to address this problem predominantly in the language domain [22, 63, 59, 88, 67, 2, 57, 28, 87, 106]. One notable milestone driving progress in this field recently is the development of the SCAN benchmark [63] posing a sequence-to-sequence translation task. The authors have demonstrated that RNNs fail catastrophically at test time when presented with a *compositionally novel* input, i.e., an unknown composition of known concepts.

However, the problem is even more elusive when it comes to imagining the dynamical implications of a visual observation, a problem referred to in this work as the *compositional or systematic visual imagination*. One reason for this is that, unlike language, images do not naturally provide such a token-based compositional representation, making the problem significantly more challenging. To tackle this problem, one must not only learn how to utilize compositional representations for systematic composition but also obtain such representations from unstructured, complex, high-dimensional pixels. Another reason is the lack of an appropriate benchmark to directly address it. Although several prior works have explored related problems, none have tackled it head-on.

Specifically, the prior research related to systematic visual imagination can be grouped into three categories. The first group [83, 105] tackles this issue by providing language alongside images, e.g., Visual Question Answering (VQA) [54, 6, 16, 5, 38], thus pursuing an indirect task that bypasses the challenge of learning compositional representation by leveraging the inherent token-based compositional nature of language. The second group [85, 13, 46, 116, 74, 109] also focuses on an indirect task, i.e., obtaining disentangled representations. The underlying hypothesis is that a disentangled representation such as those from variational autoencoders [60, 44] or object-centric representations [68, 17, 66, 53, 107, 89, 91, 90, 86] will naturally lead to a solution to the systematic generalization problem as well. However, it restricts exploring potential solutions not relying on disentanglement [108], and recent studies suggest that disentanglement may not necessarily lead to systematic generalization [74, 109]. The third group focuses on visual reasoning [19, 103, 80, 114, 4] while ignoring the problem of visual perceptual systematic generalization, essential for zero-shot problem-solving. For instance, the two closest visual reasoning benchmarks to ours, ARC [19] and Sort-of-ARC [4], exhibit these limitations. ARC lacks suitability for evaluating systematic generalization due to its non-procedural generation and lack of access to underlying component factors and rules. Sort-of-ARC, on the other hand, emphasizes inferring the underlying rule composition from a few-shot support set while ignoring the zero-shot systematic visual imagination ability—a cornerstone of our proposed benchmark.

In this paper, we introduce a new benchmark called the *Systematic Visual Imagination Benchmark* (SVIB), the first benchmark to rigorously address the compositional visual imagination problem. The SVIB poses the task as a minimal world modeling problem: one-step image-to-image generation. The objective is to generate the subsequent scene image from the current one. The underlying world dynamics operate as a relational function of object-centric factors. The benchmark presents a subset of possible combinations of the visual primitives during training, and during testing, exclusively presents compositionally novel input images to assess a model's systematic imagination ability.

SVIB offers numerous benefits for studying the compositional visual imagination problem. Our proposed task framework provides a means to jointly optimize for systematic perception and imagination, instead of optimizing an indirect objective such as disentanglement or object segmentation quality. Unlike many works in disentanglement literature focusing on a single-object scene [46, 74, 109, 98], SVIB incorporates multi-object scenes where each object is constructed via composition rules in terms of well-defined factors like color, shape, and size. SVIB also provides various difficulty levels, accommodating visual complexity from simplistic 2D images to realistically textured 3D scene images, and varying complexity of the world dynamics. Additionally, SVIB provides control over the fraction of possible factor combinations used in training, thus serving as a systematic generalization difficulty meter. The simplicity of one-step generation allows us to concentrate on the systematic imagination ability, sidestepping the computational demands of long-range generation abilities that video generation tasks often entail.

Our paper makes two main contributions: First, we propose the first benchmark to foster research on the systematic or compositional visual imagination problem, providing several unique benefits as outlined above. Second, we conduct comprehensive empirical evaluations of various baseline models on SVIB, shedding light on the current state-of-the-art capabilities in this field.

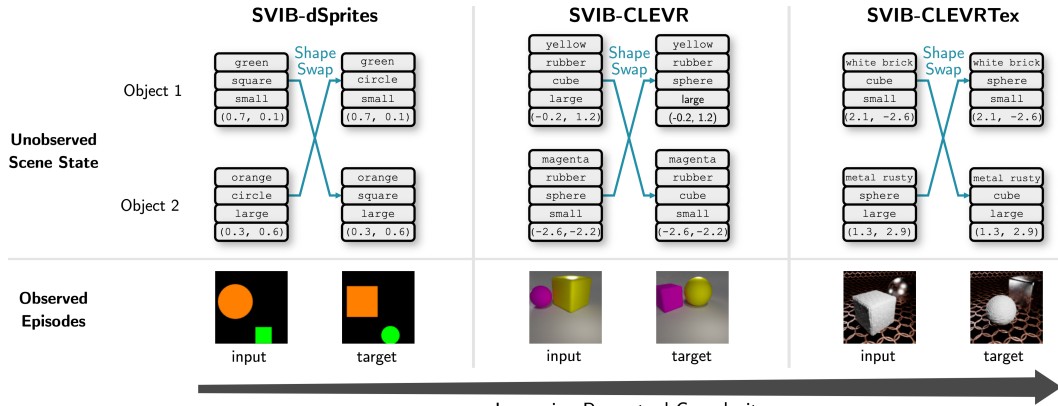

Figure 1: **Compositional Visual World in SVIB**: Our benchmark provides 2-frame videos of an underlying compositional visual world. The scenes contain two objects, each composed using intra-object factors such as color, shape, size, material, etc. In this figure, we show episodes following the *Shape-Swap* rule i.e., the input state is transformed to the target state by swapping the shapes of the two objects. We illustrate episodes from three tasks having different visual complexity levels.

## 2 SVIB: Systematic Visual Imagination Benchmark

In this section, we describe the *Systematic Visual Imagination Benchmark* or *SVIB* that we propose. The benchmark is designed to train models to perform visual imagination and measure their systematic generalization ability. Our benchmark provides a catalog of 12 *visual imagination tasks*[5]. Each *task* contains 64K episodes for training a model and 8K systematically out-of-distribution episodes for testing the model. Each episode is a two-frame video as illustrated in Fig. 1. We denote the two frames as the input image $\mathbf{x}$ and the target image $\mathbf{y}$. Using the training episodes, a model must learn to map the input image $\mathbf{x}$ to the target image $\mathbf{y}$, with a focus on accurately predicting the target images for the systematically out-of-distribution input images in the testing episodes.

### 2.1 Compositional Visual World

For each task, the episodes are generated from an underlying compositional visual world defined by *1)* a *visual vocabulary*—a library of visual primitives that act as building blocks for constructing the scenes, and *2)* a *rule* that determines the mapping from the input scene to the target scene.

**Scene Compositionality and Visual Vocabulary.** In SVIB, each scene is composed of two objects. The objects are composed of intra-object *factors* such as color, shape, size, *etc.* These factors take their values from a collection of visual primitives called a *visual vocabulary*. For example, if an object can be described by its color, shape, and size then we can define a visual vocabulary containing colors {green, blue, magenta, orange}, shapes {circle, triangle, square, star}, and sizes {tiny, small, medium, large}. Under a given vocabulary, we refer to any composition of factors that completely describes an object's appearance as a *combination*. Given our example vocabulary, two possible combinations are: blue-circle-tiny and orange-triangle-large. By placing the objects defined by these combinations inside a scene, we can construct an example scene containing a blue-circle-tiny at position $(0.1, 0.4)$ and a orange-triangle-large at position $(0.7, 0.1)$.

**Rule.** A *rule* is a function that transforms the input scene state to the target scene state. In SVIB, the rule is fixed for all episodes within a task. The rule modifies specific factor values of both objects in the scene. The rule is symmetric, i.e., the same rule is executed on both objects. An example of a rule in SVIB is the *Shape-Swap* rule that swaps the shapes of the two objects to transform the input scene state to the target scene state. In SVIB, we identify two axes of rule complexity and our tasks span these two complexity axes. The first axis is *the number of modified factors per object*. We call a rule that modifies only a single factor per object as a *single* rule. If a rule modifies multiple factors per object, we call it a *multiple* rule. The second axis is *the number of parents* i.e., the number of factors that determine the modified factor value. If only one parent is involved per factor, we call the rule an

---

[5]See our project page at https://systematic-visual-imagination.github.io.

| Rule Category | Rule Description | SVIB-dSprites | | SVIB-CLEVR | | SVIB-CLEVRTex | |
|---|---|---|---|---|---|---|---|
| | | input | target | input | target | input | target |
| **Single Atomic** | `other[shape]` ⟶ `self[shape]` | | | | | | |
| **Single Non-Atomic** | `other[shape]`, `self[shape]` ⊕⟶ `self[shape]` | | | | | | |
| **Multiple Atomic** | `self[shape]` ⟶ `self[color]`
`other[color]` ⟶ `self[size]` | | | | | | |
| **Multiple Non-Atomic** | `self[shape]`, `self[quadrant]` ⊕⟶ `self[color]`
`other[color]`, `self[quadrant]` ⊕⟶ `self[size]` | | | | | | |

Figure 2: **SVIB Tasks and the Transformation Rules.** In this illustration, we show sample episodes of all 12 tasks in our benchmark. For each instance, we show the input and target images. We describe the underlying rule that governs the transformation of the input image to the target image. In the rule description diagrams, each factor value is considered to be an integer index in its corresponding vocabulary. A direct arrow indicates a simple assignment operation and a '⊕' symbol indicates a summation operation. All assignments are performed modulo the vocabulary size of the target factor. As the rules are applied symmetrically to both objects, we describe the rules with respect to the `self` object while `other` denotes the other object.

*atomic* rule else we call it a *non-atomic* rule. With these two axes, we define 4 rule categories with increasing complexity: *single atomic*, *multiple atomic*, *single non-atomic*, and *multiple non-atomic*. For example, the aforementioned Shape-Swap rule can be categorized as *single atomic* because only one factor (i.e., shape) is modified per object whose new value depends on only one parent, i.e., the shape of the other object.

## 2.2 Systematic Training and Testing Splits

In this section, we describe how we construct the training and the testing episodes of a task. As a benchmark for studying systematic generalization, our task episodes satisfy the following three conditions: *1)* each primitive in the visual vocabulary is shown individually in the training episodes, *2)* a subset of combinations is reserved for testing and *3)* the fraction of combinations exposed during training can be controlled, offering a control knob to adjust the difficulty of generalization. To satisfy these, we proceed as follows.

**Core Combinations and Testing Combinations.** We first construct a set of *core combinations*—the smallest set that contains each primitive in the visual vocabulary at least once. For instance, for our previous example of a visual vocabulary, we can define the set of core combinations as follows: {`green-circle-tiny`, `blue-square-small`, `magenta-square-medium`, `orange-star-large`}. Next, from all the remaining combinations of primitives, we reserve 20% of the combinations as *testing combinations*. For more details, see Appendix D.2.

**The $\alpha$-Rating and Training Combinations.** Each training split in our benchmark is associated with an $\alpha$-*rating*. To create a training split having a specific $\alpha$-rating, we randomly select an $\alpha$ fraction of the combinations that remain after setting aside the core and the testing combinations. These selected combinations are then added to the core combinations to obtain the set of *training combinations*. The $\alpha$-rating acts as a control knob over how difficult it is to generalize for a model trained on this split. A higher $\alpha$-rating corresponds to exposing more combinations in the training split, thereby providing an easier generalization setting. Similarly, a low $\alpha$-rating corresponds to a more difficult generalization setting. In Figure 3, we provide an illustration.

**Generating the Episodes.** To sample an episode given the training combinations, we randomly sample two combinations and position the objects randomly to create an input scene. Next, the task

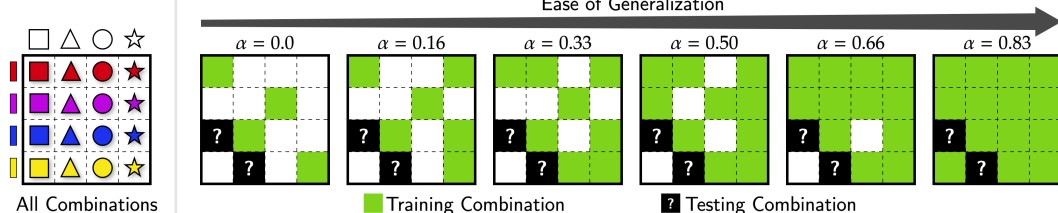

Figure 3: **The $\alpha$-Rating of Training Splits in SVIB.** *Left:* In this illustration, we showcase a library of 4 shape primitives and 4 color primitives that result in a total of 16 possible combinations or objects. *Right:* In training episodes, we present each shape and color primitive individually, however, we do not show all possible combinations. To control the ease of solving the generalization task, we define an $\alpha$-rating of the training split, the fraction of the combinations presented during training. A higher $\alpha$-rating corresponds to more combinations shown in the training split. In testing episodes, we present only the held-out combinations.

rule is applied to the input scene to obtain the target scene. The input and the target scenes are rendered to generate the two frames of an episode. By sampling episodes repeatedly, we obtain the complete training split. The testing combinations are used to generate the testing split analogously.

### 2.3 Contents of the Benchmark

**Tasks.** In SVIB, we provide 12 tasks organized into 3 subsets based on the visual complexity of their underlying visual world: *1)* SVIB-dSprites is based on visually simple 2D scenes; *2)* SVIB-CLEVR is based on visually simple 3D scenes; and *3)* SVIB-CLEVRTex is based on 3D scenes with complex textures. This gradation in visual complexity is illustrated in Figure 1. Within each of these three subsets, we provide 4 tasks corresponding to 4 rules with increasing rule complexity: *Single Atomic* (S-A), *Single Non-Atomic* (S-NA), *Multiple Atomic* (M-A), and *Multiple Non-Atomic* (M-NA). These 4 rules are illustrated in Figure 2. For their formal definitions, see Appendix D.1.1. For each rule, we provide 3 training splits with distinct $\alpha$-ratings denoted as *Easy* split ($\alpha = 0.6$), *Medium* split ($\alpha = 0.4$) and *Hard* split ($\alpha = 0.2$) and a common testing split. Each training split contains 64K episodes and each testing split contains 8K episodes. For each episode, we provide the input image, the target image, the ground-truth scene descriptions, and the ground-truth object masks. In Figure 6, we provide the complete directory structure of the benchmark.

**Omni-Composition Datasets.** In SVIB, we provide an *omni-composition dataset* for each of the 3 visual worlds i.e., SVIB-dSprites, SVIB-CLEVR, and SVIB-CLEVRTex. An omni-composition dataset is a dataset containing unpaired images that capture all possible combinations of primitives under the visual vocabulary of its visual world. The use of omni-composition datasets for solving the benchmark tasks is optional.

## 3 Metrics

**MSE.** A natural metric for measuring the systematic generalization performance of a model is the mean squared error (MSE) between predicted and target images on the testing set episodes (denoted as $\text{MSE}^{\text{OOD}}$). We can also compute how much worse the OOD MSE is relative to the in-distribution MSE via the *systematic generalization gap*: $\log \text{MSE}^{\text{OOD}} - \log \text{MSE}^{\text{ID}}$, where $\text{MSE}^{\text{ID}}$ denotes the in-distribution MSE.

**LPIPS.** A key limitation of MSE is that it can overly focus on low-level errors, e.g., small deviations that do not significantly impact human perception. Therefore we recommend reporting LPIPS [115], an error metric shown to better correspond to human judgments. LPIPS works by taking two images as input, extracting their features via a pre-trained image classification model, and computing a weighted distance between the extracted features. In our experiments, we use the official implementation[6].

---

[6] https://github.com/richzhang/PerceptualSimilarity

# 4 Baselines

In this section, we describe the baselines that we evaluate in our experiments. These were selected considering two main implications of our benchmark: evaluating systematic generalization in vision models and world models. We evaluate two categories of baselines: *1)* Image-to-Image Models, and *2)* State-Space Models. Additionally, we evaluate an *Oracle* model to obtain the best-case performance and to set a milestone of success on the benchmark tasks.

## 4.1 Image-to-Image Models

In this category, we consider end-to-end neural network models that take the input image $\mathbf{x}$ and try to predict the target image $\mathbf{y}$. Within the model, the input image is mapped to an intermediate hidden representation $\mathbf{e}$ which is used to generate the prediction $\hat{\mathbf{y}}$. This can be described as follows:

$$\mathbf{e} = \text{Encoder}_\theta(\mathbf{x}) \qquad \Longrightarrow \qquad \hat{\mathbf{y}} = \text{Decoder}_\gamma(\mathbf{e})$$

We test two variants that capture the two extremes of how expressive the intermediate representation $\mathbf{e}$ is. The first variant employs a CNN as the encoder to map the input image to single-vector representation $\mathbf{e}$, thus creating a bottleneck of low expressiveness. We denote this baseline as *Image-to-Image CNN* or I2I-CNN. The second variant employs a Vision Transformer or ViT as the encoder and produces a representation that is a collection of multiple vectors—as many as the patches in the image [25]. We denote this baseline as *Image-to-Image ViT* or I2I-ViT.

**Oracle.** To set a milestone of success on our benchmark, we construct a baseline where instead of employing an image encoder to obtain the representation $\mathbf{e}$, we directly stack the vector representations of the ground truth scene factors and provide it to the decoder. For more details, see Appendix E.3.

## 4.2 State-Space Models

State-Space Models (or SSMs) are a category of models that first encode the input image $\mathbf{x}$ to a latent representation $\mathbf{z_x}$ and then apply a latent-level dynamics model to predict the target scene latent.

$$\mathbf{z_x} = \text{Encoder}_\phi(\mathbf{x}) \qquad \Longrightarrow \qquad \hat{\mathbf{z}}_\mathbf{y} = \text{Dynamics}_\theta(\mathbf{z_x}) \qquad \Longrightarrow \qquad \hat{\mathbf{y}} = \text{Probe}_\gamma(\hat{\mathbf{z}}_\mathbf{y}),$$

where $\mathbf{z_x}, \hat{\mathbf{z}}_\mathbf{y} \in \mathcal{Z}$. We test two variants that employ two different kinds of latent structures. The first variant adopts Variational Auto-Encoding or VAE for obtaining latent scene representations—a popular representation method for world modeling [60, 42, 41]. The encoded latent is a single vector where factors of scene variation are expected to be disentangled and captured by individual dimensions of the vector. We denote this variant as *SSM-VAE*. The second variant adopts Slot Attention Video or SAVi—a popular object-centric representation method [68, 61, 107]. SAVi represents each frame by a set of $N$ object vectors, also known as slots. Furthermore, the slot order is preserved across frames, facilitating the training of a slot-level dynamics model. We denote this variant as *SSM-Slot*. Finally, since it is not possible to compute the MSE or LPIPS performance from the predicted latent, we train a probe $\text{Probe}_\gamma(\cdot)$ that takes the predicted latent $\hat{\mathbf{z}}_\mathbf{y}$ and decodes it to generate the target image $\hat{\mathbf{y}}$. To ensure a fair performance comparison, we keep the probe's architecture to be the same as that of the decoder used in the Image-to-Image baselines and the Oracle. For more details, see Appendix E.4.

# 5 Related Work

**Systematic Generalization in Language.** In the language domain, a substantial number of studies, including SCAN [63], have examined systematic generalization through sequence-to-sequence tasks that translate different sequence forms [22, 59, 88, 67, 2, 57, 28, 87, 106]. Some other benchmarks like gSCAN [83] and ReaSCAN [105] utilize additional inputs such as images and videos alongside language [81, 92, 69]. In *Visual Question Answering* or VQA, studies test systematicity by providing images with unseen object combinations [54, 6] or presenting questions in unseen formats [16, 5, 38]. However, all of these either test systematic generalization in language or utilize language inputs when testing visual compositionality.

**Representation Learning.** Another group of works questions specific representation learning methodologies and whether they support systematic generalization. One line of representation methods seeks disentangled single-vector representations [44, 58, 62]. However, studies that investigate it either do

not quantify (e.g., via $\alpha$-rating) the degree to which OOD inputs systematically differ from the training [98, 109] or tackle only visually simplistic and single-object scenes [98, 109, 97, 74, 73], unlike ours. Furthermore, [45], relying on RL, lacks the simplicity of our one-step prediction framework. Higgins *et al.* [46] test the importance of disentanglement in a symbol-to-image task, however, the learning is partly supervised by symbol labels. Zhao *et al.* [116] study generative models (e.g., VAE) and whether the learned distributions capture OOD regions of the observation space, however, they lack a focus on perception unlike ours. Another line of representation methods seeks object-centric representations [35, 27, 66, 34, 26, 68, 89, 86]. Although these are considered promising to support OOD generalization [36, 90], very few studies investigate this potential. Such studies tend to lack well-defined and systematic factor recombination for OOD testing [23, 24, 111, 70, 102] or rely on RL [24, 111, 70, 102], lacking the simplicity of one-step prediction.

**Visual Reasoning.** In the visual domain, several studies test reasoning abilities by inferring the underlying rule from a support set and then finding images from a candidate set that conform to that rule [10, 113, 48, 14, 47, 96, 49]. Alternatively, there are odd-one-out tasks that find the rule-violating image from an input image set without requiring a separate support set [112, 71] and tasks that categorize input images into several groups [29, 39, 99, 75, 52]. However, their learning setup is discriminative and not generative, unlike ours. Some reasoning tasks require learning to generate [19, 103, 80, 114], however, these tasks fall outside the scope of visual systematic generalization. Assouel *et al.* [4] improve upon the ARC benchmark [19] by properly defining elementary primitives and procedurally generating the dataset to make it suitable to study systematic generalization. However, the primitives and their combinations are concerned with scene dynamics and not the visual observation itself, unlike ours. Furthermore, both benchmarks [19, 4] use coarse and simplistic observations, whereas ours provides visually complex high-resolution images.

# 6   Experiments

We test the five baselines as outlined in Section 4: Image-to-Image CNN, Image-to-Image ViT, SSM-VAE, SSM-Slot, and Oracle on the 12 benchmark tasks outlined in Section 2.3. In Figures 4, 10, and 11, we report the LPIPS, the MSE, and the systematic generalization gaps, respectively. Since LPIPS better reflects the human judgment of qualitative performance, we base our conclusions in this section on LPIPS. In our analysis, we consider Oracle's performance as a yardstick for task success. In Fig. 5 and 14, we show samples of the model predictions.

## 6.1   Effect of $\alpha$ on Systematic Generalization

**Performance at $\alpha = 0.0$.** When $\alpha = 0.0$, the training split only includes the core combinations. In all tasks, we note that all baselines fail completely for $\alpha = 0.0$. This is supported by the qualitative results in Fig. 5, where most predicted images are garbled blobs. This can be expected since at $\alpha = 0.0$, the factors are highly correlated in the training split, making it difficult for the learner to infer the true causal parent in the rule. For instance, in the Shape-Swap task in SVIB-dSprites: the learner sees only the core combinations: {`green-circle-tiny`, `blue-square-small`, `magenta-square-medium`, `orange-star-large`} where every shape value is always shown with a specific color value and a specific size value, making it impossible (even for the Oracle) to determine whether the true causal parent is color, shape, or size.

**Performance with Increasing $\alpha$.** As $\alpha$ is increased from 0.0, the difficulty of systematic generalization becomes less severe and we see a consistent downward slope in the error plots across baselines and tasks. This is also supported by the qualitative results in Fig. 5, where, with increasing $\alpha$, a greater number of baselines are able to improve their predictions. Since training splits with larger $\alpha$ expose more combinations, there is more opportunity for the baselines to overcome learning spurious correlations and discover the compositional primitives and the causal rule to apply.

**Performance at $\alpha = 0.6$.** Our easiest training split corresponds to $\alpha = 0.6$ which exposes the largest fraction of combinations during training. Yet, on most tasks, most baselines are not able to reach the performance of Oracle—although it does happen in some select cases. From this, we can say that our benchmark is largely unsolved with significant room for progress.

**Performance of Oracle.** We note that Oracle is the best performer as it is the only baseline that can solve the tasks as early as at $\alpha = 0.2$. In comparison to Oracle, other baselines show a much flatter

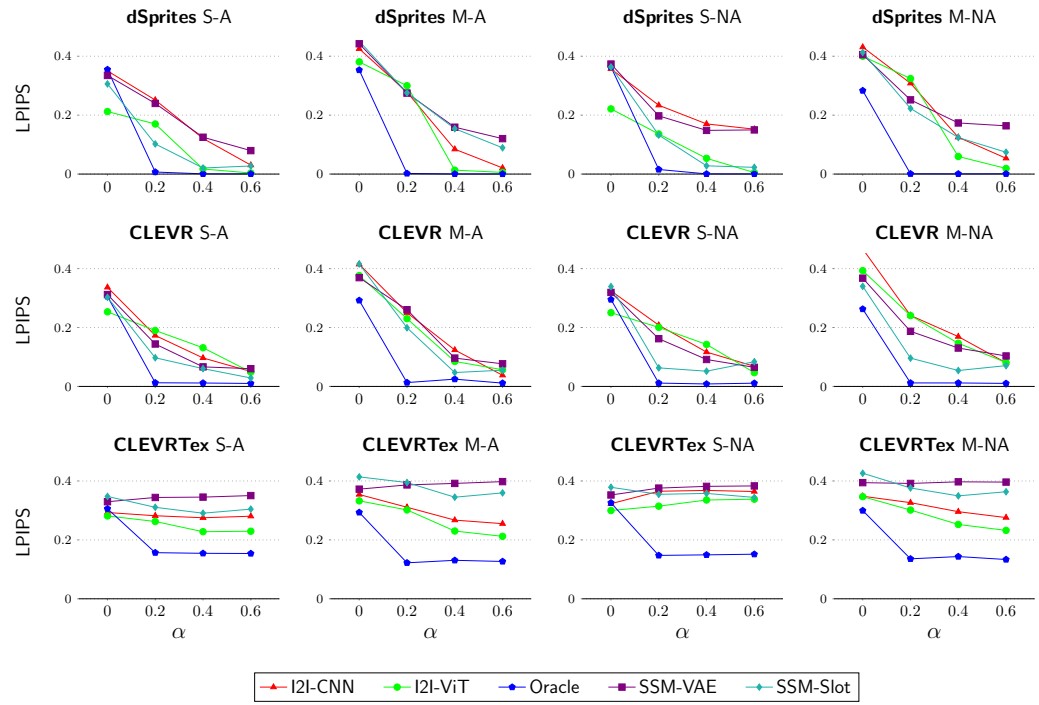

Figure 4: **Systematic Generalization in Visual Imagination.** We plot LPIPS with respect to $\alpha$ for all benchmark tasks and baselines on the systematically out-of-distribution (OOD) test set. Lower is better. For SVIB-dSprites, SVIB-CLEVR, and SVIB-CLEVRTex, we evaluate the tasks: single atomic (S-A), single non-atomic (S-NA), multiple atomic (M-A), and multiple non-atomic (M-NA).

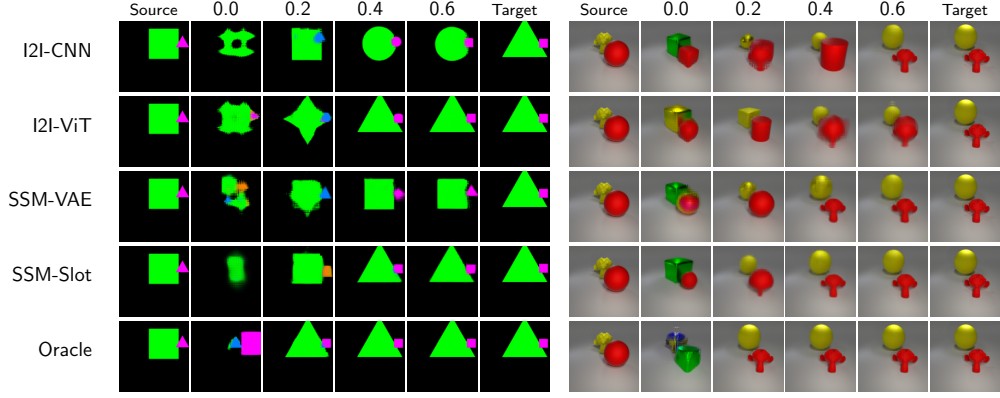

Figure 5: **Qualitative Performance of the baselines on SVIB-dSprites and SVIB-CLEVR.** We illustrate the predictions made by various baselines on the Shape-Swap task, where the task is to swap the shapes of the two objects in the input image. We show predictions on a test episode for various baselines for varying $\alpha$ of the training split.

downward slope. This can be explained by considering that SVIB requires systematic perception i.e., learning to perceive the input in terms of useful tokens—explicitly or implicitly—while also generalizing systematically to OOD inputs. Oracle is designed with access to the ground-truth tokens directly and it does not need to learn systematic perception via training. On the contrary, other baselines carry the additional burden of learning systematic perception along with rule learning—contributing to their worse performance.

**Performance on SVIB-CLEVRTex.** The performance of baselines on SVIB-CLEVRTex requires a special mention as we see that the curves with respect to $\alpha$ are almost flat and the slope is not

Table 1: **LPIPS Performance on SVIB.** This table summarizes the LPIPS performance results of various models evaluated across all 12 tasks of SVIB. The values are computed by averaging the results from four tasks for each difficulty level.

| Models | SVIB-dSprites | | | SVIB-CLEVR | | | SVIB-CLEVRTex | | |
|---|---|---|---|---|---|---|---|---|---|
| | Easy | Medium | Hard | Easy | Medium | Hard | Easy | Medium | Hard |
| I2I-CNN | 0.0643 | 0.1251 | 0.2678 | 0.0593 | 0.1265 | 0.2181 | 0.2938 | 0.3014 | 0.3210 |
| I2I-ViT | **0.0084** | **0.0359** | 0.2326 | **0.0583** | 0.1263 | 0.2153 | **0.2529** | **0.2614** | **0.2950** |
| SSM-VAE | 0.1284 | 0.1515 | 0.2410 | 0.0763 | 0.0961 | 0.1884 | 0.3819 | 0.3789 | 0.3745 |
| SSM-Slot | 0.0534 | 0.0817 | **0.1834** | 0.0596 | **0.0533** | **0.1139** | 0.3429 | 0.3357 | 0.3590 |
| Oracle | 0.0006 | 0.0006 | 0.0064 | 0.0107 | 0.0141 | 0.0122 | 0.1414 | 0.1444 | 0.1404 |

as large as that in the other simpler environments i.e., SVIB-dSprites and SVIB-CLEVR. That is, the baselines struggle to systematically generalize in SVIB-CLEVRTex. A notable exception is Oracle which can solve the tasks starting at $\alpha = 0.2$. What distinguishes Oracle from other baselines is its systematic perception ability. As such, we can attribute the degraded performance of other baselines in SVIB-CLEVRTex to their poor systematic perception. This shows that higher visual complexity can make systematic perception more difficult in comparison to visually simple scenes of SVIB-dSprites and SVIB-CLEVR. This observation is also supported by the predictions visualized in Fig. 14. It is noteworthy that our proposed benchmark provides a unique opportunity to study systematic imagination in visually complex scenes as the existing image-to-image benchmarks provide visually toy-like scenes [19, 4].

## 6.2   Comparison of Baselines

We now compare the baselines in more detail. We draw our conclusions based on tasks with ratings of $\alpha = 0.2$ and $0.4$ since the tasks are too difficult for $\alpha = 0.0$ and, in select cases, too easy for $\alpha = 0.6$.

**Image-to-Image Models.** In Image-to-Image models, we compare two encoders: CNN and ViT. In SVIB-dSprites, we note that ViT is generally superior to CNN. This is also evidenced in the qualitative results of SVIB-dSprites in Fig. 5 where ViT can correctly predict for $\alpha \geq 0.4$ while CNN continues to predict the wrong object shapes on all $\alpha$ values. It is somewhat surprising that ViT, which lacks any explicit design to encourage systematic generalization, can generalize at all. However, in SVIB-CLEVR and SVIB-CLEVRTex, the performances of CNN and ViT are comparable. This suggests that although ViT is more capable than CNN on SVIB-dSprites, the greater visual complexity of SVIB-CLEVR and SVIB-CLEVRTex can make ViT struggle similarly to CNN.

**State-Space Models.** Comparing SSM-VAE and SSM-Slot, we find that the SSM-Slot generally outperforms the SSM-VAE. This is further evidenced by the qualitative results of SVIB-dSprites in Fig. 5 where SSM-VAE predicts an incorrect shape even at $\alpha = 0.6$ while SSM-Slot is able to predict correctly for $\alpha \geq 0.4$. However, both SSMs still remain significantly worse than the Oracle.

**State-Space Modeling versus Image-to-Image Modeling.** We compare our best-performing image-to-image model with our best-performing state-space model i.e., I2I-ViT versus SSM-Slot. In SVIB-dSprites, both models perform comparably. We believe this is due to the visual simplicity of SVIB-dSprites which allows ViT to learn systematic perception implicitly without any explicit inductive biases such as those imposed on the SSMs. In SVIB-CLEVR, SSM-Slot seems to generalize significantly better than I2I-ViT. This is also evidenced in the qualitative results in Fig. 5, where SSM-Slot can correctly generate the red monkey for $\alpha \geq 0.4$ while I2I-ViT fails for all $\alpha$ values. We hypothesize that since SSMs enforce an inductive bias on the latent, this can eventually help in learning systematic perception. In SVIB-CLEVRTex, both baselines fail completely. In fact, SSM-Slot fails worse than I2I-ViT. We think this is due to a familiar problem of the conventional Slot Attention-based models that they cannot extract objects properly in visually complex scenes [55, 91].

### 6.3 Summary of Experiment Results

We summarize the main experimental findings and pointers for making progress on our benchmark:

1. All baselines fail completely at $\alpha = 0.0$. Although most baselines demonstrate improved performance with increasing $\alpha$, only a select few can eventually solve some of the tasks and reach the performance of the Oracle. This highlights the challenging nature of our tasks and indicates a substantial opportunity for improvement.

2. Systematic perception is central to solving our tasks. However, visual complexity can make perception and, as a result, task-solving much more difficult. Our benchmark provides a unique opportunity to study systematic generalization along the axis of visual complexity.

3. Excluding the Oracle, in SVIB-dSprites and SVIB-CLEVR, ViT outperforms all other baselines on the *Easy* and *Medium* training splits but loses to the object-centric baseline SSM-Slot on the *Hard* training split. In SVIB-CLEVRTex, ViT outperforms all other baselines but significantly falls short of reaching the successful performance of the Oracle.

4. Models based on single-vector representations (e.g., I2I-CNN and SSM-VAE) tend to perform worse than their multi-vector counterparts (e.g., I2I-ViT and SSM-Slot).

## 7 Conclusion

We introduced the *Systematic Visual Imagination Benchmark* (SVIB)—a novel image-to-image generation benchmark that focuses on visual compositionality and systematic generalization. SVIB is procedurally generated and contains multi-object scenes, with each object compositionally constructed from well-defined visual primitives. The underlying image-to-image mapping rule is designed to require extracting the underlying visual factors. Our experiments showed that our benchmark is yet to be solved and highlights an important limitation of the current models. We also showed that systematic perception and visual complexity are important aspects of this problem. We hope that our benchmark will enable the development of more capable world modeling and perception approaches.

**Limitations and Future Extensions.** Our benchmark makes several simplifying choices to support faster model development and ease of analysis. However, this leaves several limitations and avenues for extending the benchmark. First, from a world modeling perspective, a richer benchmark may be constructed by introducing action-conditioning, stochasticity, longer episodes, occlusions, 3D viewpoints, *etc.* Second, our dynamics are fixed within a task. Our benchmark can be extended to have distinct dynamics in each episode constructed compositionally using dynamical primitives. Third, our benchmark can be extended by introducing greater realism e.g., greater visual complexity, more objects, more primitives, *etc.* Fourth, our dynamics involve symmetric rules and a future extension may introduce non-symmetric rules. Another avenue is to increase emphasis on relational abstractions. Lastly, future works may also consider investigating whether large pre-trained models (e.g., [43]) can provide necessary priors for systematic generalization.

## Acknowledgements and Disclosure of Funding

This work is supported by Brain Pool Plus Program (No. 2021H1D3A2A03103645) through the National Research Foundation of Korea (NRF) funded by the Ministry of Science and ICT.

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
