# A  Datasheet for Datasets

## A.1  Motivation

### Question 1: For what purpose was the dataset created?

The dataset was created as a test-bed to evaluate the systematic generalization ability of visual imagination models, with an emphasis on compositionality at the level of intra-object factors (e.g., color, shape, size, etc.) and visual complexity.

### Question 2: Who created the dataset (e.g., which team or research group) and on behalf of which entity (e.g., company, institution, organization)?

The dataset was created by the authors who are affiliated with Machine Learning and Mind Lab (MLML) situated in the School of Computing at Korea Advanced Institute of Science and Technology (KAIST) and Rutgers University.

### Question 3: Who funded the creation of the dataset?

This work is supported by Brain Pool Plus Program (No. 2021H1D3A2A03103645) through the National Research Foundation of Korea (NRF) funded by the Ministry of Science and ICT.

## A.2  Composition

### Question 1: What do the instances that comprise the dataset represent (e.g., documents, photos, people, countries)?

The dataset comprises several tasks. Within each task, we provide 1) several training splits of different difficulties of systematic generalization, and 2) a test split. Within each split, we provide several pairs of images, with the first image in each pair serving as the input and the second image serving as the prediction target. These images are purely synthetic and procedurally generated using the Blender[7] and the Spriteworld API[8]. The images showcase multi-object scenes where the objects have simple shapes (e.g., sphere, cube), simple colors (e.g., red, blue), standard materials (e.g., brick, rubber, metal), and multiple sizes (e.g., small, medium, large). No real data (e.g., about people or countries) was collected or used in our data creation process. Along with the images, we also provide scene meta-data files detailing the configurations of the objects, lighting, camera, and background of the scene. We also provide the ground truth object masks.

### Question 2: How many instances are there in total of each type?

In our dataset, there are a total of 12 tasks, 4 tasks per environment (SVIB-dSprites, SVIB-CLEVR, and SVIB-CLEVRTex). Within each task, we provide 4 training splits corresponding to $\alpha$ values $0.0$, $0.2$, $0.4$, and $0.6$. The training splits $0.2$, $0.4$, and $0.6$ are denoted as Hard, Medium, and Easy, respectively. Within each training split, we provide 64000 input and target images. Furthermore, within a task, we provide a test split containing 8000 out-of-distribution input and target images. We illustrate the complete directory structure of our dataset in Figure 6.

### Question 3: Does the dataset contain all possible instances or is it a sample (not necessarily random) of instances from a larger set?

For each intra-object factor (e.g., color or shape), we define a library of primitives from which the factor takes values. These libraries of primitives are finite meaning that these libraries are not an exhaustive list of all possible values that a factor can theoretically take. Nevertheless, our libraries are designed to capture all standard factor values as also done in the previous works [54, 55].

---

[7] https://www.blender.org
[8] https://github.com/deepmind/spriteworld

Table 2: Factor primitives for various object properties in SVIB-dSprites.

| Shape | Color (RGB) | Size |
|-------|-------------|------|
| circle | (0, 255, 0) | 0.125 |
| triangle | (255, 0, 255) | 0.225 |
| square | (0, 127, 255) | 0.325 |
| star_4 | (255, 127, 0) | 0.425 |

Table 3: Factor primitives for various object properties in SVIB-CLEVR.

| Shape | Color (RGB) | Size | Material |
|-------|-------------|------|----------|
| SmoothCube_v2 | (255, 0, 0) | 1.0 | Rubber |
| Sphere | (0, 255, 0) | 1.5 | MyMetal |
| SmoothCylinder | (0, 0, 255) | 2.0 | |
| Suzanne | (0, 255, 255) | | |
| | (255, 0, 255) | | |
| | (255, 255, 0) | | |

**Question 4: What data does each instance consist of?**

Each instance consists of 4 files: `source.png`, `source.json`, `target.png`, `target.json`. The `source.png` and `target.png` are PNG files containing the $128 \times 128$-sized source and target images, respectively. The `source.json` and `target.json` are JSON files detailing the scene meta-data of the source and the target scenes, respectively. See Figure 7 for an illustration of the JSON contents.

**Question 5: Is there a label or target associated with each instance?**

Yes, in each instance, the prediction target is a $128 \times 128$ RGB image (denoted as the target image) and is provided as the PNG file `target.png`.

**Question 6: Is any information missing from individual instances?**

No, our scenes are not partially observable as we took care to ensure that all objects in our scenes have at least a certain number of visible pixels in the observation images `source.png` and `target.png`.

**Question 7: Are relationships between individual instances made explicit (e.g., users' movie ratings, social network links)?**

No relationships are present between individual instances.

**Question 8: Are there recommended data splits (e.g., training, development/validation, testing)?**

Yes, for each task, we provide 4 training splits and one OOD test split. The 4 training splits capture different levels of systematic generalization difficulty and correspond to $\alpha$ values 0.0, 0.2, 0.4, and 0.6. We do not provide a separate validation split, it is completely up to the learner how they want to leverage the training splits for validation e.g., via hold-out validation, $K$-fold cross-validation, leave-one-out validation, etc.

**Question 9: Are there any errors, sources of noise, or redundancies in the dataset?**

No.

**Question 10: Is the dataset self-contained, or does it link to or otherwise rely on external resources (e.g., websites, tweets, other datasets)?**

Yes, the dataset is self-contained and does not require external resources to work with.

Table 4: Factor primitives for various object properties in SVIB-CLEVRTex. For materials, we use 8 free textures provided by Poliigon (`https://www.poliigon.com`)

| Shape | Size | Material |
|---|---|---|
| Cone | 1.0 | PoliigonBricksFlemishRed001 |
| Cube | 1.5 | PoliigonBricksPaintedWhite001 |
| Cylinder | 2.0 | PoliigonChainmailCopperRoundedThin001 |
| Suzanne | | PoliigonFabricDenim003 |
| Icosahedron | | PoliigonFabricFleece001 |
| NewellTeapot | | PoliigonMetalSpottyDiscoloration001 |
| Sphere | | PoliigonRoofTilesTerracotta004 |
| Torus | | PoliigonWoodFlooring061 |

**Question 11: Does the dataset contain data that might be considered confidential (e.g., data that is protected by legal privilege or by doctor-patient confidentiality, data that includes the content of individuals' non-public communications)?**

No.

### A.3 Collection Process

**Question 1: How was the data associated with each instance acquired?**

The data was procedurally generated using the Spriteworld and Blender APIs.

**Question 2: What mechanisms or procedures were used to collect the data (e.g., hardware apparatus or sensor, manual human curation, software program, software API)?**

The SVIB-dSprites images were generated from the Spriteworld API, the SVIB-CLEVR images using Blender 2.78, and the SVIB-CLEVRTex using Blender 2.93. All implementations were done in Python. The Blender processes were run on GPU instead of CPU-only for faster rendering. In the creation of individual splits such as specific training or testing splits, a single instance of a modern Nvidia GPU was enough, demanding less than 25GB of GPU memory.

**Question 3: If the dataset is a sample from a larger set, what was the sampling strategy (e.g., deterministic, probabilistic with specific sampling probabilities)?**

To generate a specific split e.g., a specific training or testing split, the combinations of factor values that would be exposed in that split were predefined as discussed in Section 2. From this predefined set of combinations, a specific combination was selected uniformly at random to instantiate each object within the scene. If a generated scene had an object that was fully occluded by another, or if two objects were too close, such scenes were discarded.

**Question 4: Who was involved in the data collection process (e.g., students, crowdworkers, contractors) and how were they compensated (e.g., how much were crowdworkers paid)?**

The authors of this paper alone were involved.

**Question 5: Over what timeframe was the data collected?**

The data generation process took several months and involved multiple refinement and deliberation steps.

**Question 6: Were any ethical review processes conducted (e.g., by an institutional review board)?**

No.

## A.4 Preprocessing / Cleaning / Labeling

**Question 1: Was any preprocessing/cleaning/labeling of the data done (e.g., discretization or bucketing, tokenization, part-of-speech tagging, SIFT feature extraction, removal of instances, processing of missing values)?**

No. Since our data was procedurally generated, there was no raw data collected or used. As such, there was nothing to preprocess or clean.

**Question 2: Was the raw data saved in addition to the preprocessed/cleaned/labeled data (e.g., to support unanticipated future uses)?**

Not Applicable.

**Question 3: Is the software that was used to preprocess or clean or label the data available?**

Not Applicable.

## A.5 Uses

**Question 1: Has the dataset been used for any tasks already?**

Yes, the dataset has been used in our paper to evaluate various state-of-the-art architectures in terms of their ability to systematically generalize.

**Question 2: Is there a repository that links to any or all papers or systems that use the dataset?**

We plan to list these on the official website of this benchmark.

**Question 3: What (other) tasks could the dataset be used for?**

Given that we provide scene metadata and object masks, future explorations might investigate the utility of mask supervision in solving our tasks. That said, we also note that using the scene metadata and object masks is not the intended path to solving our benchmark.

**Question 4: Is there anything about the composition of the dataset or the way it was collected and preprocessed/cleaned/labeled that might impact future uses?**

No, the dataset was procedurally generated. No real data about people was used. As such, it is unlikely that using our data would cause direct harm. While the intended goal of our benchmark is to spur the development of more capable models, future deployments of such models should be mindful of any potential harms.

**Question 5: Are there any tasks for which the dataset should not be used?**

We are not aware of tasks that, if performed via our dataset, would lead to direct negative consequences.

## A.6 Distribution

**Question 1: Will the dataset be distributed to third parties outside of the entity (e.g., company, institution, organization) on behalf of which the dataset was created?**

Yes, the dataset will be publicly available on our official website.

**Question 2: How will the dataset be distributed (e.g., tarball on website, API, Github)?**

The dataset will be distributed via the official project page `https://systematic-visual-imagination.github.io/`.

**Question 3: When will the dataset be distributed?**

The full dataset will be distributed upon acceptance of the paper. For the purposes of the review, a smaller sample of the dataset containing 100 instances per split is available now.

**Question 4: Will the dataset be distributed under a copyright or other intellectual property (IP) license, and/or under applicable terms of use (ToU)?**

The dataset will be released under the most liberal Creative Commons license i.e., CC0. While not mandatory, we do encourage future works to cite our paper when using our benchmark.

**Question 5: Have any third parties imposed IP-based or other restrictions on the data associated with the instances?**

We are not aware of any IP-based restrictions imposed by third parties on our dataset.

**Question 6: Do any export controls or other regulatory restrictions apply to the dataset or to individual instances?**

We are not aware of any.

## A.7 Maintenance

**Question 1: Who is supporting/hosting/maintaining the dataset?**

The research lab at KAIST led by Sungjin Ahn will maintain the dataset.

**Question 2: How can the owner/curator/manager of the dataset be contacted (e.g., email address)?**

The manager can be contacted via email: `sungjin.ahn@kaist.ac.kr`

**Question 3: Is there an erratum?**

If errors are found at a later date, we will provide an erratum on the official website.

**Question 4: Will the dataset be updated (e.g., to correct labeling errors, add new instances, delete instances)?**

Yes, in case there are updates, we will release the new version on our website. The older versions will also remain available.

**Question 5: If the dataset relates to people, are there applicable limits on the retention of the data associated with the instances (e.g., were individuals in question told that their data would be retained for a fixed period of time and then deleted)?**

Not Applicable.

**Question 6: Will older versions of the dataset continue to be supported/hosted/maintained?**

Yes.

**Question 7: If others want to extend/augment/build on or contribute to the dataset, is there a mechanism for them to do so?**

Yes, we shall release all our source code, including the code we used to generate the datasets under a highly permissive CC0 license. Others can freely download, modify and create their own variants of the dataset.

## A.8 Author Statement of Responsibility

We, the authors, confirm that we bear all responsibility in the case of violation of rights and licenses.

Table 5: **Accuracy of Ground-Truth Factor Prediction in SVIB-CLEVRTex**. We perform a factor prediction task where we take SVIB-CLEVRTex images as input and predict the factors for the scene. We indicate these by $\text{Shape}_i$, $\text{Size}_i$, $\text{Mat}_i$, where $i = 1, 2$ denotes object index in the scene based on closeness to the camera. We also predict the background material denoted with $\text{Mat}_{bg}$.

| Test Accuracy | $\text{Shape}_1$ | $\text{Shape}_2$ | $\text{Size}_1$ | $\text{Size}_2$ | $\text{Mat}_1$ | $\text{Mat}_2$ | $\text{Mat}_{bg}$ |
|---|---|---|---|---|---|---|---|
| Top-1 Acc. (%) | 92.81 | 88.39 | 94.99 | 93.46 | 95.67 | 92.98 | 99.99 |

# B    Additional Experiment Results

In this section, we provide additional experimental results that could not be included in the main paper due to the limitation of space.

## B.1    MSE on the Benchmark Tasks

In Figure 10, we report the in-distribution and out-of-distribution MSE for all benchmark tasks.

## B.2    Generalization Gap on the Benchmark Tasks

In Figure 11, we report the systematic generalization gap for all benchmark tasks. The systematic generalization gap is defined as the difference between the OOD MSE and in-distribution MSE in the log scale.

## B.3    Qualitative Results

In Figures 12, 13 and 14, we visualize the predicted images of various baselines on the benchmark tasks.

## B.4    MSE on the Analysis Tasks

In Figure 15, we plot the MSE performance on the analysis tasks. A detailed description of how the analysis tasks are constructed is provided in Section D.1.2.

## B.5    Comparison of In-Distribution Performance between Image-to-Image and State-Space Models

In Figures 4 and 10, one observation is that state-space models (SSMs) generally have worse in-distribution MSE than image-to-image models. This can be expected since the image-to-image models minimize the prediction error directly in the image space while state-space models minimize the prediction error in the latent space which is an indirect objective.

## B.6    Factor Prediction

The SVIB-CLEVRTex environment, which is highly textured data, has a relatively high level of visual complexity. Consequently, within this environment, the importance of visual recognition abilities can significantly increase. Therefore, we conducted an experiment in which we trained a simple 4-layer CNN encoder to take SVIB-CLEVRTex images as input and predict GT labels to ascertain whether $128 \times 128$ image resolution can provide sufficient information in this setting.

# C    Additional Related Work

**Benchmarks for Video Prediction.** In the realm of video prediction, there are several real-world benchmarks [33, 50, 20, 21]. However, these cannot be used to study systematic visual imagination since, being real-world datasets, do not provide control over factor combinations and their demarcation between training and testing. Although synthetic benchmarks provide more control over such data-generating factors: [11, 94, 37, 110, 82, 9, 65], such benchmarks have only so far focused on in-distribution video prediction and do not focus on systematic out-of-distribution evaluation.

Table 6: **Limitations of Existing Studies:** The table contrasts our proposed Systematic Visual Imagination Benchmark (SVIB) with the existing studies. We note that existing studies do not offer a benchmark for evaluating systematic perception ability in the image domain.

| Study | | Task Modality | Systematic Perception | Perceptual Complexity |
|---|---|---|---|---|
| Language | SCAN [63] | Text → Text | Not Applicable | Not Applicable |
| | gSCAN [83] | Image + Text → Text | ✓ | ▲ (Toy Multi-Object) |
| Disentanglement | Xu *et al.* [109] | Image → Latent | ✓ | ▲ (Toy Single-Object) |
| | Montero *et al.* [74] | Image → Latent | ✓ | ▲ (Toy Single-Object) |
| Visual Reasoning | ARC [19] | Image → Image | ✗ (Non-Systematic) | ✗ (Very Low Resolution) |
| | Sort-of-ARC [4] | Image → Image | ✗ (In-Distribution) | ✗ (Very Low Resolution) |
| **SVIB (Ours)** | | Image → Image | ✓ | ✓ (Realistic Multi-Object) |

**Benchmarks for Generalization in RL.** Although OGRE and NovPhy [1, 32] provide a platform to study out-of-distribution objects, these lack well-defined primitives and systematic evaluation unlike ours. [56] focuses on causal induction rather than systematic perception unlike ours. [31] also provides a platform to study out-of-distribution environments, however, its scene dynamics involve interactions between low-level powder particles rather than high-level abstractions, unlike ours. Yet, all of these are RL benchmarks meaning that they do not provide an isolated way of studying the world modeling itself which is possible in our benchmark.

**Large-Scale Pretraining and Prompting.** Large-scale pretraining of vision models followed by prompting has shown remarkable zero-shot abilities. Some of these are focused on performing classification or class-conditional image generation via prompting [51, 104] unlike ours which focuses on image-to-image generation tasks. A more recent line focuses on image-to-image prediction via prompting [100, 8, 101, 93]. A large-scale pre-trained image-to-image model can indeed have capabilities as a world model given appropriate in-context prompts. However, since large-scale pretraining is done on real datasets without access to the underlying factors, it is difficult to quantify how well the pre-trained model generalizes. On the other hand, in our benchmark, the underlying factors are known and generalization ability can be clearly assessed in terms of $\alpha$ required for task success. Furthermore, large-scale pretraining is an expensive endeavor and which makes it difficult to quickly test and analyze models. On the other hand, with our simple and lightweight benchmark, it is possible to test a model with less than a single modern GPU within a 2-day training timeframe. There have been some studies on the compositional generalization ability of large models, however, to our knowledge, these are confined to the text domain [64, 3].

# D   Details of the Benchmark

## D.1   Rule Definitions

In this section, we will provide a pseudo-code for all the rules proposed in this benchmark. We first describe notations that we use in our pseudo-code and then proceed to describe the rules.

**Symmetricity.** Our rules are designed to be applied symmetrically to both objects. As such, we describe the rule definitions relative to just one of the objects that we denote as `self`. We will denote the other object as `other`. We will access a factor of an object using square brackets e.g., `other['color']` shall denote the color of the `other` object.

**Integer Factor Value.** In the following descriptions, we will assume that the value of a factor is represented as an integer. For instance, the color of an object `self['color']` can take a value in $0, 1, \ldots,$ `num_colors` $- 1$, where `num_colors` is the total number of color primitives in the color vocabulary. Similarly, the shape of an object `self['shape']` can take a value in $0, 1, \ldots,$ `num_shapes` $- 1$, where `num_shapes` is the total number of shape primitives in the shape vocabulary.

## D.1.1   Benchmark Rules

We define four benchmark tasks for each subset: SVIB-dSprites, SVIB-CLEVR, and SVIB-CLEVRTex. These tasks align with the four rule categories outlined in Section 2.

**Single Atomic.** In this task, the transformation is executed by swapping the shapes of the two objects in the input image. For brevity, we sometimes call it the *Shape-Swap* task and denote it as S-A.

$$\text{self['shape']} \leftarrow \text{other['shape']}$$

**Single Non-Atomic.** This task involves a transformation where the shape of each object is updated based on the shapes of both objects in the input image, as determined by a lookup table. For brevity, we sometimes denote this task as S-NA.

$$\text{self['shape']} \leftarrow (\text{self['shape']} + \text{other['shape']}) \mod \text{num\_shapes}$$

**Multiple Atomic.** In this task, the transformation involves simultaneous updates to the color and size of each object. The new color is determined by the object's own shape, and the new size is determined by the color of the other object. Both determinations are made by a lookup table. For brevity, we sometimes denote this task as M-A.

$$\text{self['color']} \leftarrow \text{self['shape']} \mod \text{num\_colors}$$
$$\text{self['size']} \leftarrow \text{other['color']} \mod \text{num\_sizes}$$

**Multiple Non-Atomic.** In this task, the transformation involves simultaneous updates to the color and size of each object. The new color is determined by the object's own shape and quadrant, and the new size is determined by the other object's color and quadrant. Both determinations are made by a lookup table. For brevity, we sometimes denote this task as M-NA.

$$\text{self['color']} \leftarrow (\text{self['shape']}$$
$$+ \text{quadrant(self['position'])}) \mod \text{num\_colors}$$
$$\text{self['size']} \leftarrow (\text{other['color']}$$
$$+ \text{quadrant(self['position'])}) \mod \text{num\_sizes}$$

For SVIB-CLEVRTex, the color factor indicates texture.

### D.1.2   Analysis Tasks

We design 16 analysis tasks, a broad pool of tasks from which we eventually choose the rules to construct the final benchmark. The analysis rules in these tasks are designed to be broad-based by keeping the following points in mind:

1. All rules taken together should cover all factors.
2. All rules taken together should cover various types of factor interactions: no interaction, interactions between factors of the same object, and interactions between factors of different objects.

With these design considerations we design the rules as shown in Figure 16. For this, we first create four atomic rules. We then construct rules with greater complexity by incrementally adding factors and parent edges to the causal graph. By analyzing the model performance on these rules in the dSprites environment, we choose 4 rules—one per each rule complexity category—that will be used to create the final benchmark tasks. The rules for the final benchmark tasks are chosen by identifying the rules for which Oracle can solve the tasks well while the other baselines struggle. This is done to balance the difficulty and solvability of our benchmark. The fact that Oracle can solve the tasks shows that our tasks are indeed solvable and do not pose an insurmountable challenge to the community. At the same time, we also ensure that the tasks we choose are difficult to solve for the current models. Note that it is expensive to run 5 baselines on 16 tasks (totaling 80 experiments) for all environments. Therefore, we perform the analysis experiments only in the dSprites environment. The 4 finalized rules are then used for SVIB-CLEVR and SVIB-CLEVRTex as well. When adopting the finalized rules to SVIB-CLEVRTex, we interpret the color factor of the SVIB-dSprites environment as the texture factor of the SVIB-CLEVRTex environment.

### D.2   Core Combinations

To show each primitive in the visual vocabularies individually during training, we create a set of core combinations. This is the smallest set that contains all visual primitives at least once. Given the factor vocabularies $\mathcal{V}_1, \ldots, \mathcal{V}_M$, we construct the core combinations as described in Algorithm 1.

---

**Algorithm 1** Collecting Core Combinations

---

1: **Input:** Vocabularies $\mathcal{V}_1, \ldots, \mathcal{V}_M$
2: **Output:** Set $\mathcal{C}$ containing core combinations, each combination represented as tuple.
3:
4: **Initialize** an empty set $\mathcal{C}$
5: $N \leftarrow \max_{m=1}^{M} |\mathcal{V}_m|$
6:
7: **for** $i = 0$ **to** $N - 1$ **do**
8:     **Initialize** an empty tuple $T$
9:     **for** $m = 1$ **to** $M$ **do**
10:         **Append** $\mathcal{V}_{m,i \bmod |\mathcal{V}_m|}$ to $T$
11:     **end for**
12:     **Add** $T$ to $\mathcal{C}$
13: **end for**
14:
15: **return** $\mathcal{C}$

---

# E    Details of the Baselines

In this section, we provide the details of the implementation of the baselines. In Table 7, we report the hyperparameter for modules used in our baselines.

## E.1    Image-to-Image Models

The image-to-image models consist of an encoder and a decoder.

$$\mathbf{e} = \text{Encoder}_\theta(\mathbf{x}) \qquad \Longrightarrow \qquad \hat{\mathbf{y}} = \text{Decoder}_\gamma(\mathbf{e})$$

We implement two variants for the encoder: CNN and ViT. To implement the decoder, we adopt a transformer decoder. The complete model is trained in an end-to-end manner by minimizing the mean squared error (MSE) between the predicted image and the target image, i.e., $\mathcal{L}(\theta, \gamma) = ||\hat{\mathbf{y}} - \mathbf{y}||^2$.

## E.2    State-Space Models

To train the modules of the state-space models, we adopt the following three-stage approach:

1. In the first stage, we train the encoder $\text{Encoder}_\phi$ network. For this, we use the combined dataset of input and target images of a given task.

2. In the second stage, we freeze the $\text{Encoder}_\phi$ and train only the dynamics model $\text{Dynamics}_\theta$ via a simple latent-level MSE loss: $\mathcal{L}(\theta) = ||\text{Dynamics}_\theta(\mathbf{z_x}) - \mathbf{z_y}||^2$. This dynamics model is implemented as a 4-layer transformer.

3. In the third stage, we freeze the encoder $\phi$ and the dynamics model $\theta$ and train a *probe* parametrized by $\gamma$. The probe takes the latent $\mathbf{z_y}$ predicted by the dynamics model and decodes it to render the target image $\hat{\mathbf{y}}$. We deliberately implement the probe as a transformer decoder using the same implementation that was also used for the Image-to-Image models and the Oracle to perform a fair comparison of performance across baselines.

### E.2.1    SSM-VAE

We train the VAE on the task images using its standard auto-encoding objective. After training the VAE, we use the mean of the posterior network as the $\text{Encoder}_\phi(\cdot)$ to acquire latent representations of task images. Given the latent representations, we train a dynamics model to predict the target latent given the input latent by minimizing a latent-level MSE objective.

### E.2.2    SSM-Slot

We train SAVi in a fully unsupervised manner, by considering input and target images $(\mathbf{x}, \mathbf{y})$ from a task as a 2-frame video. After training the SAVi, we utilize the encoder portion to obtain the aligned

Table 7: **Baseline Hyperparameters.** In this table, we provide the hyperparameters for all baselines evaluated in our experiments.

| Module | Hyperparameter | Benchmark Subset | | |
|---|---|---|---|---|
| | | SVIB-dSprites | SVIB-CLEVR | SVIB-CLEVRTex |
| General | Batch Size | 32 | 40 | 40 |
| | Training Steps | 160K | 160K | 160K |
| CNN Encoder | Kernel Size | 5 | 5 | 5 |
| | Stride | 2 | 2 | 2 |
| | Padding | 2 | 2 | 2 |
| | Hidden Size | 64 | 64 | 64 |
| | Learning Rate | 0.0001 | 0.0001 | 0.0001 |
| ViT Encoder | # Encoder Blocks | 8 | 8 | 8 |
| | # Encoder Heads | 8 | 8 | 8 |
| | Hidden Size | 192 | 192 | 192 |
| | Dropout | 0.1 | 0.1 | 0.1 |
| | Learning Rate | 0.0001 | 0.0001 | 0.0001 |
| SSM-VAE | # Dynamic Blocks | 4 | 4 | 4 |
| | VAE Latents | 64 | 64 | 64 |
| | VAE $\beta$ | 1.0 | 1.0 | 1.0 |
| | VAE $\sigma$ | 0.01 | 0.01 | 0.01 |
| | Learning Rate | 0.0003 | 0.0003 | 0.0003 |
| SSM-Slot | # Dynamic Blocks | 4 | 4 | 4 |
| | # Dynamic Heads | 4 | 4 | 4 |
| | # Slots | 4 | 4 | 4 |
| | Slot Size | 64 | 64 | 64 |
| | # Iterations | 3 | 3 | 3 |
| | Learning Rate | 0.0002 | 0.0002 | 0.0002 |
| Transformer Decoder | # Decoder Blocks | 8 | 8 | 8 |
| | # Decoder Heads | 4 | 4 | 4 |
| | Patch Size | $4 \times 4$ pixels | $4 \times 4$ pixels | $4 \times 4$ pixels |
| | Hidden Size | 192 | 192 | 192 |
| | Dropout | 0.1 | 0.1 | 0.1 |
| | Learning Rate | 0.0003 | 0.0003 | 0.0003 |

slot representations for the task images. Given the slot representations, we train a dynamics model to predict the slots for the target image given the slots for the input image by minimizing a slot-level MSE objective.

### E.3 Oracle

Oracle is designed to bypass the perception step by directly taking the ground truth scene factors as input. Oracle consists of an encoder that maps the ground truth scene factors to a representation comprised of a set of vectors $\mathbf{e} \in \mathbb{R}^{(NM+G) \times D}$, where $N$ is the number of objects, $M$ is the number of factors per object and $G$ is the number of global scene factors such as background texture, lighting, camera, etc. $D$ is the size of each embedding. In $\mathbf{e}$, the categorical factors are represented via learned embeddings while the float factors are represented as sine-cosine embeddings. With this design, Oracle has the perfect systematic generalization ability for perception since it always receives a correctly factored multi-vector representation as the input, regardless of what the input image is. Finally, a transformer decoder decodes $\mathbf{e}$ to predict the target image.

$$\mathbf{e} = \text{Encoder}_{\phi}(\mathbf{s_x}) \qquad \Longrightarrow \qquad \hat{\mathbf{y}} = \text{TransformerDecoder}_{\gamma}(\mathbf{e})$$

### E.4 Transformer Decoder

In previous sections, we refer to the use of a transformer decoder for 1) predicting the target image in the image-to-image models and 2) as a probe to predict the target image given the predicted target state by the SSMs. In this section, we describe the implementation of this transformer decoder.

Given an input representation, the transformer decoder predicts the patches of the target image. With $H_{\text{patch}} \times W_{\text{patch}}$ as the dimensions of a patch, an $H \times W$ image can be seen as a collection of $L = \frac{H \times W}{H_{\text{patch}} \times W_{\text{patch}}}$ patches. The transformer is given a collection of positional embeddings as input— one for each of the $L$ patches. Within the transformer, a layer of transformer performs self-attention between these patches, cross-attention on the given input representation to decode, followed by a standard MLP layer. All these steps are done with residual connections. The output of the transformer is $L$ embeddings. Each of these embeddings is mapped and reshaped using a linear layer to generate each patch of size $H_{\text{patch}} \times W_{\text{patch}} \times C$, where $C$ is the number of channels ($C = 3$ in our case for RGB images).

A similar decoder design has also been employed in models such as MAE and OSRT [43, 84]. Compared to CNN decoders which have local receptive fields, the transformer enables much longer range interactions via the attention layers [77, 18].

# F   Discussion

**Balancing Difficulty and Solvability.** Although our results show that our benchmark is unsolved, yet, while designing this benchmark, we took caution to avoid posing an excessively difficult or impossible challenge to the community. The solvability of our benchmark is confirmed by the success of Oracle on our benchmark. Such solvability is a crucial aspect of our benchmark which sets us apart from other benchmarks like ARC [19] which eventually turned out too difficult.

**Alpha Efficiency.** A model that can generalize with smaller $\alpha$ can be considered superior to a model that requires a much larger $\alpha$ value to generalize. By providing the ability to measure the required $\alpha$, our benchmark provides a novel measurable objective for making new modeling advances. For instance, although ViT can generalize on SVIB-dSprites for $\alpha \geq 0.4$, Oracle can do it with smaller $\alpha \geq 0.2$, making it superior to ViT.

# G   Impact Statement

Our work in this paper focuses on the development of a novel benchmark, designed to evaluate the systematic visual imagination capabilities of vision and machine learning models. Our benchmark uses procedurally generated scenes and does not involve sensitive personal data or human workers. While our benchmark targets the development of more robust and effective machine learning models, we urge that models tested and improved upon using our benchmark should follow strong ethical standards. They should refrain from deployment in potentially harmful use cases, such as surveillance or spreading misinformation. Despite these broader ethical considerations, our work does not pose immediate ethical concerns.

# H   Reproducibility Statement

Our experimental setup was implemented using PyTorch [78], with each experiment requiring less than 20GB of GPU memory and concluding within a two-day timeframe. The complete benchmark and the code used to generate the benchmark as well as that used to conduct experiments is made accessible via our project page: `https://systematic-visual-imagination.github.io`.

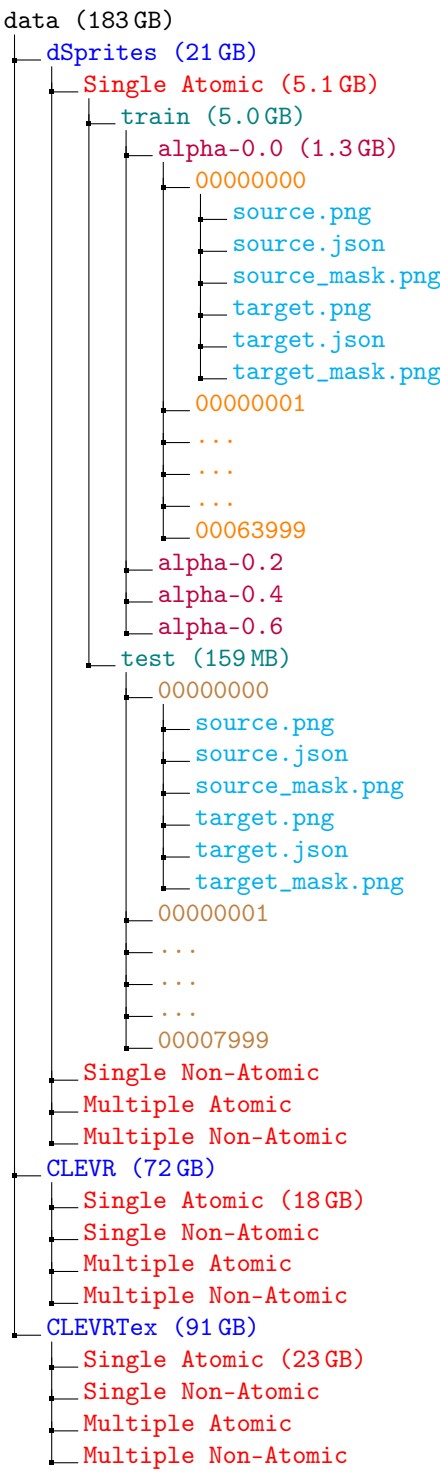

Figure 6: Directory structure of the dataset. We show a detailed view of the SVIB-dSprites directory. The structure of other environments i.e., SVIB-CLEVR and SVIB-CLEVRTex are identical. We also show the directory sizes on the disk in parentheses next to the directory names.

```
{
  "image_filename": "00000000",
  "objects": [
    {
      "shape": "triangle",
      "size": 0.325,
      "rotation": 0.0,
      "2d_coords": [0.629, 0.365],
      "color": [255, 0, 255]
    },
    {
      "shape": "circle",
      "size": 0.425,
      "rotation": 0.0,
      "2d_coords": [0.684, 0.797],
      "color": [0, 127, 255]
    }
  ]
}
```

```
{
  "image_filename": "000000110120338217",
  "objects": [
    {
      "color": [1.0, 1.0, 0.0, 1.0],
      "material": "MyMetal",
      "3d_coords": [2.924, -2.007, 2.0],
      "pixel_coords": [77, 62, 7.241],
      "size": 2.0,
      "shape": "Sphere",
      "rotation": 0.0
    },
    {
      "color": [0.0, 0.0, 1.0, 1.0],
      "material": "Rubber",
      "3d_coords": [-2.835, 1.059, 2.0],
      "pixel_coords": [50, 30, 12.723],
      "size": 2.0,
      "shape": "SmoothCylinder",
      "rotation": 0.0
    }
  ],
  "Camera": [6.990, -6.999, 5.379],
  "Lamp_Back": [-1.111, 2.506, 6.118],
  "Lamp_Key": [6.451, -3.099, 4.898],
  "Lamp_Fill": [-3.825, -3.888, 2.036]
}
```

Figure 7: Examples of JSON files describing the ground truth scene structure of an SVIB-dSprites image (left) and an SVIB-CLEVR image (right). Within each of the images, there are two objects. Each object has factors such as shape, size, rotation, 2D coordinates, 3D coordinates, and RGBA color values. In SVIB-CLEVR, we also have additional metadata specifying the poses of the camera and the lights.

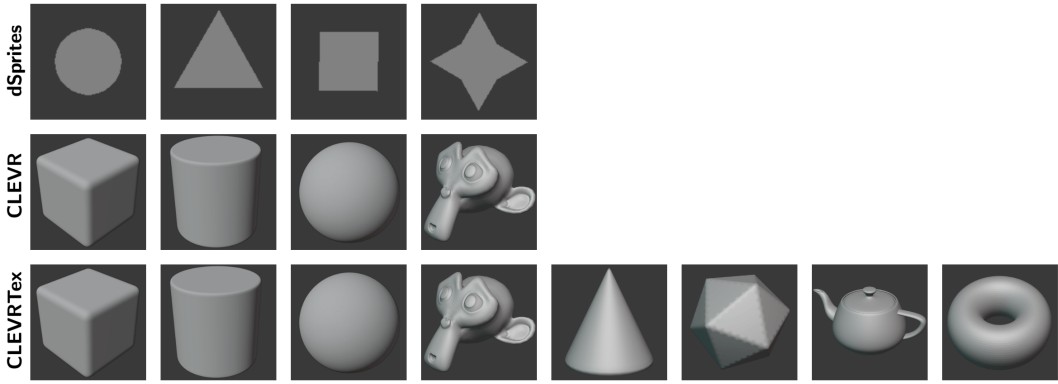

Figure 8: Illustrations of shapes used in various environments of our benchmark. For SVIB-dSprites, the shapes are generated using the Spriteworld API (https://github.com/deepmind/spriteworld). The shapes used in our benchmark for SVIB-CLEVR and SVIB-CLEVRTex are taken from the Github code (https://github.com/karazijal/clevrtex-generation) released by Karazija *et al.*[55].

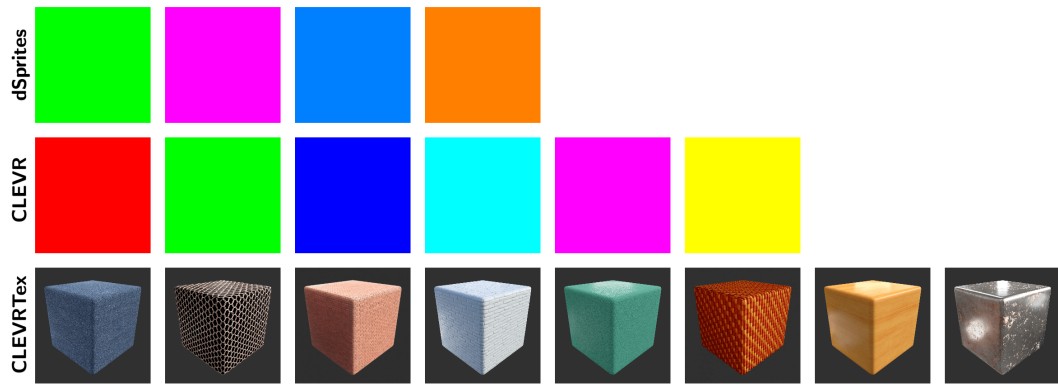

Figure 9: Illustrations of colors and materials used in various environments of our benchmark. For SVIB-CLEVRTex, we use 8 free textures provided by Poliigon (`https://www.poliigon.com`) that are also used in the original CLEVRTex data generation code (`https://github.com/karazijal/clevrtex-generation`) released by Karazija *et al.*[55].

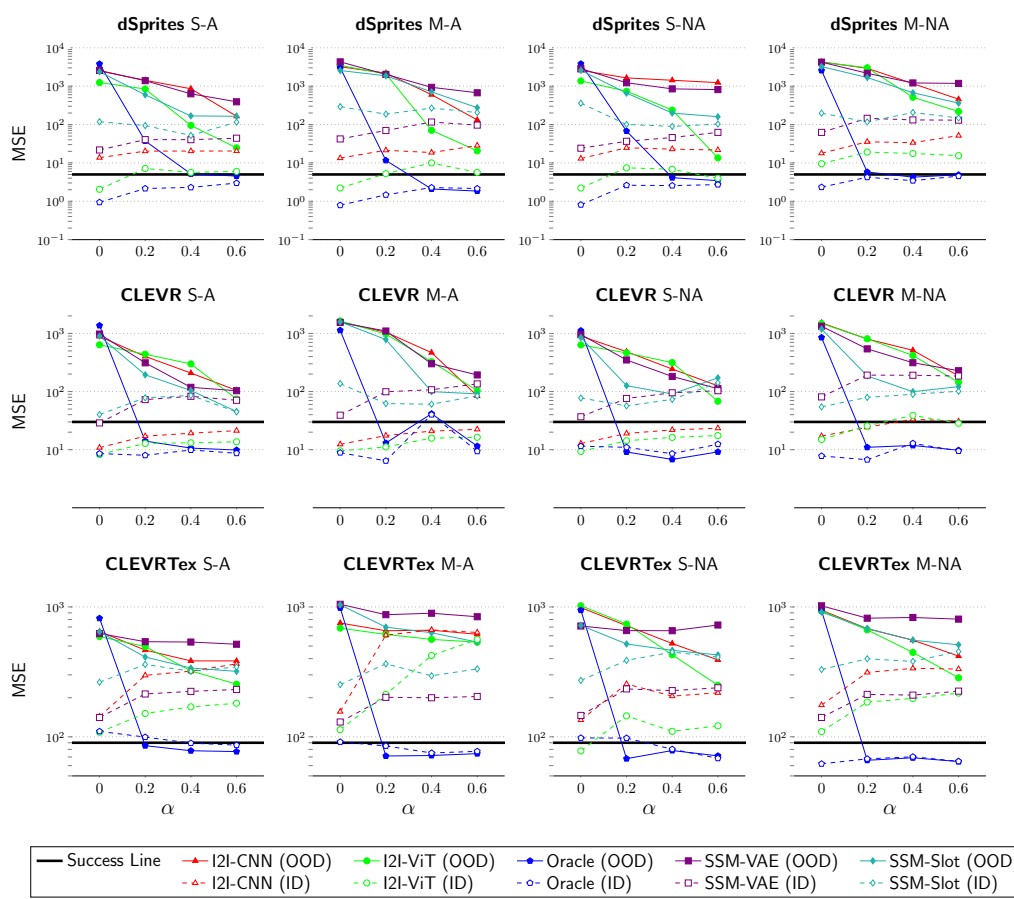

Figure 10: **MSE Performance.** We report the in-distribution and out-of-distribution MSE for all benchmark tasks.

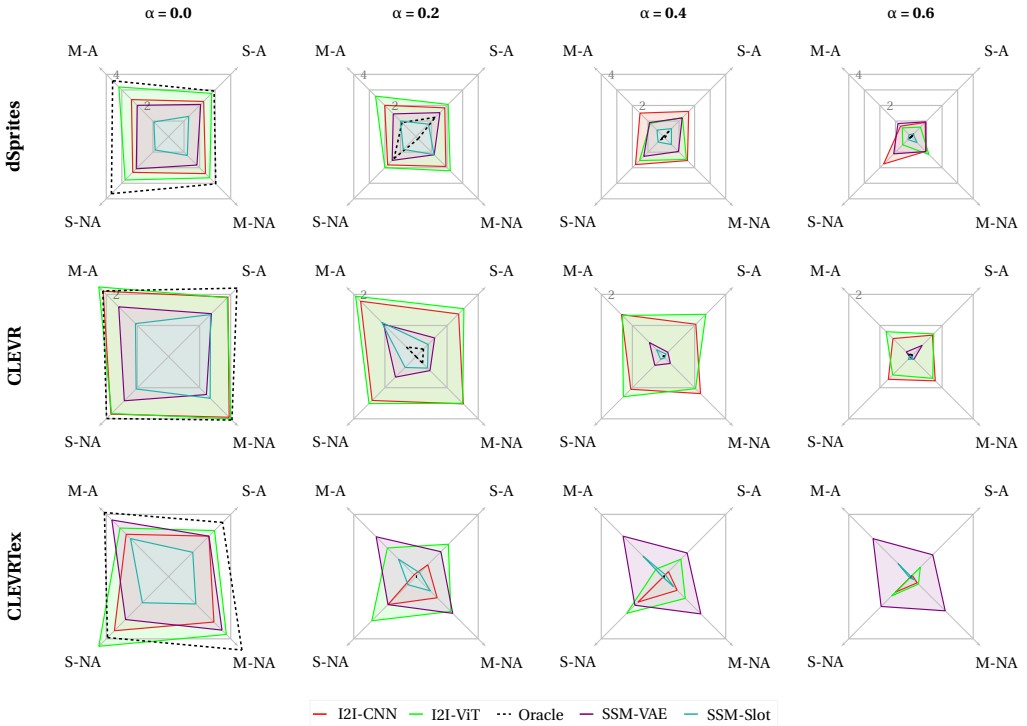

Figure 11: **Systematic Generalization Gap.** We report the systematic generalization gap for all benchmark tasks.

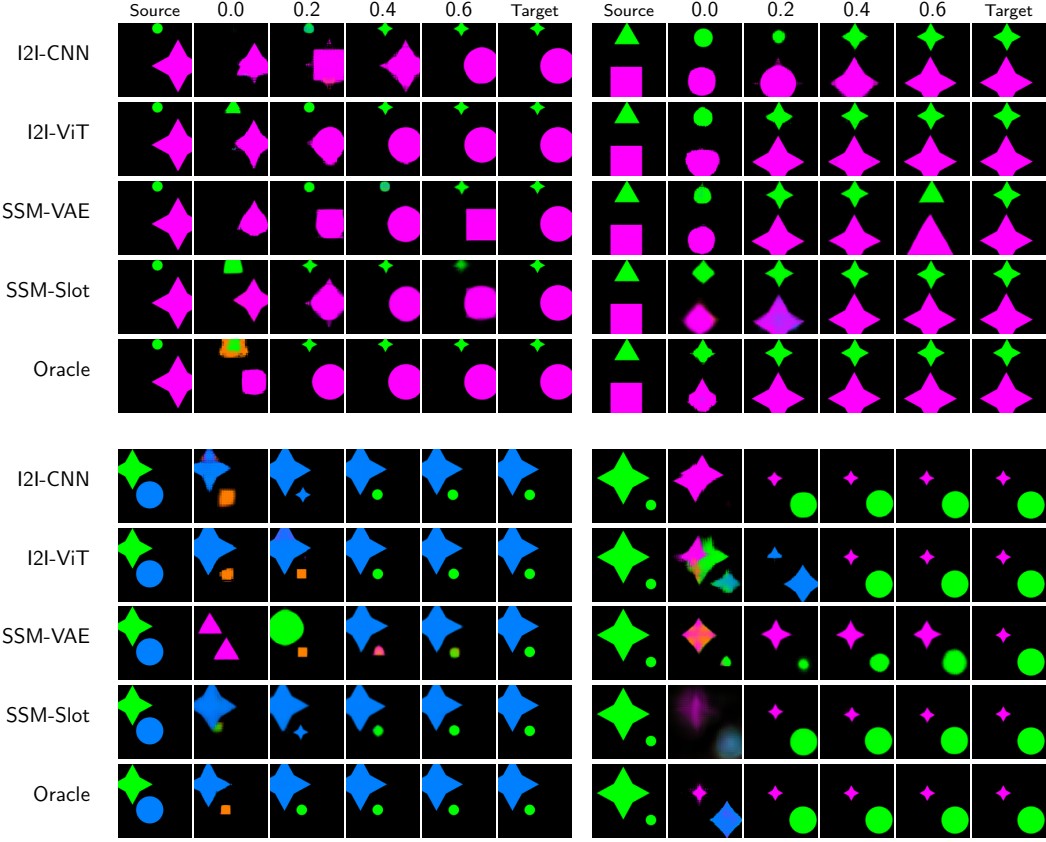

Figure 12: **Qualitative Results on Benchmark Tasks in SVIB-dSprites.** We visualize the baseline predictions on all 4 tasks. *Top-Left:* Single Atomic. *Top-Right:* Single Non-Atomic. *Bottom-Left:* Multiple Atomic. *Bottom-Right:* Multiple Non-Atomic.

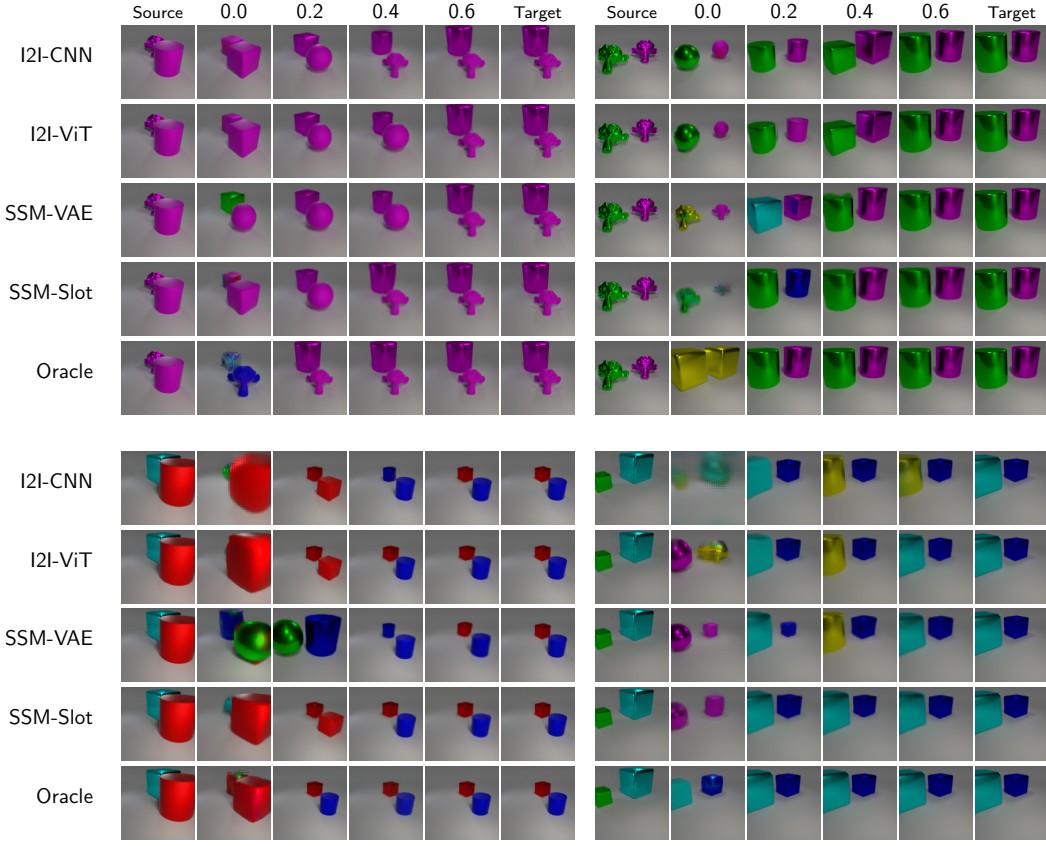

Figure 13: **Qualitative Results on Benchmark Tasks in SVIB-CLEVR.** We visualize the baseline predictions on all 4 tasks. *Top-Left:* Single Atomic. *Top-Right:* Single Non-Atomic. *Bottom-Left:* Multiple Atomic. *Bottom-Right:* Multiple Non-Atomic.

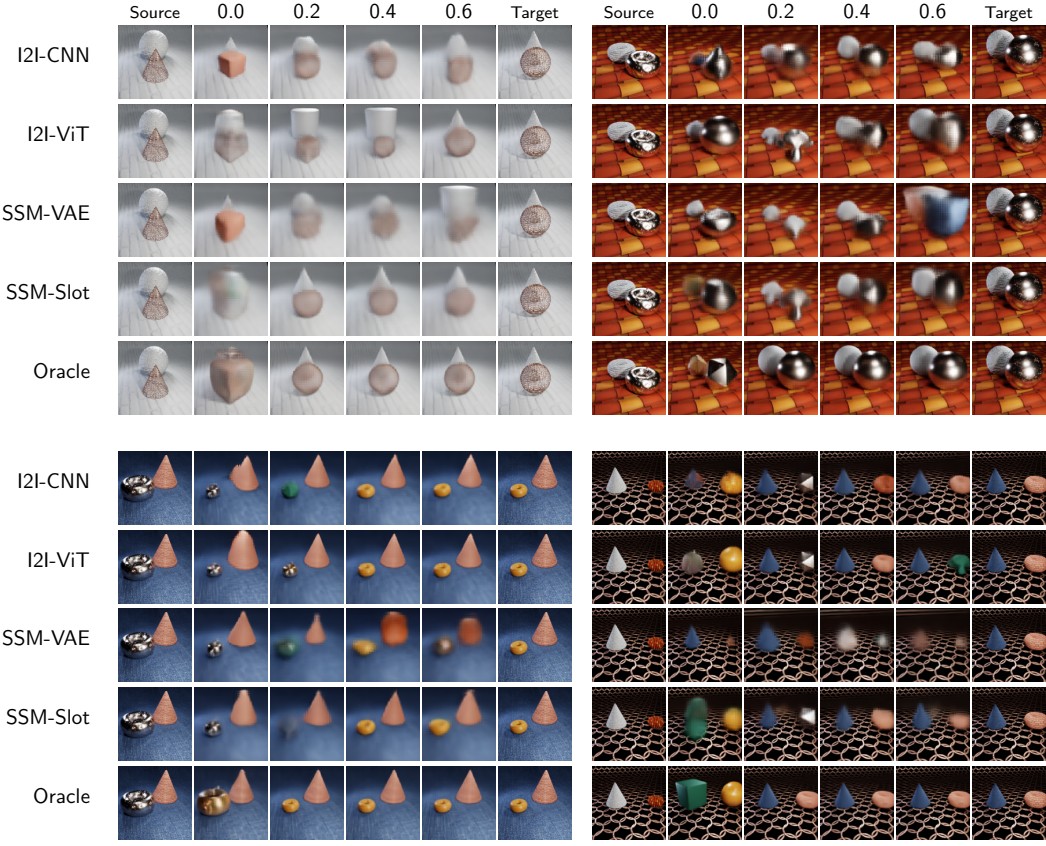

Figure 14: **Qualitative Results on Benchmark Tasks in SVIB-CLEVRTex.** We visualize the baseline predictions on all 4 tasks. *Top-Left:* Single Atomic. *Top-Right:* Single Non-Atomic. *Bottom-Left:* Multiple Atomic. *Bottom-Right:* Multiple Non-Atomic.

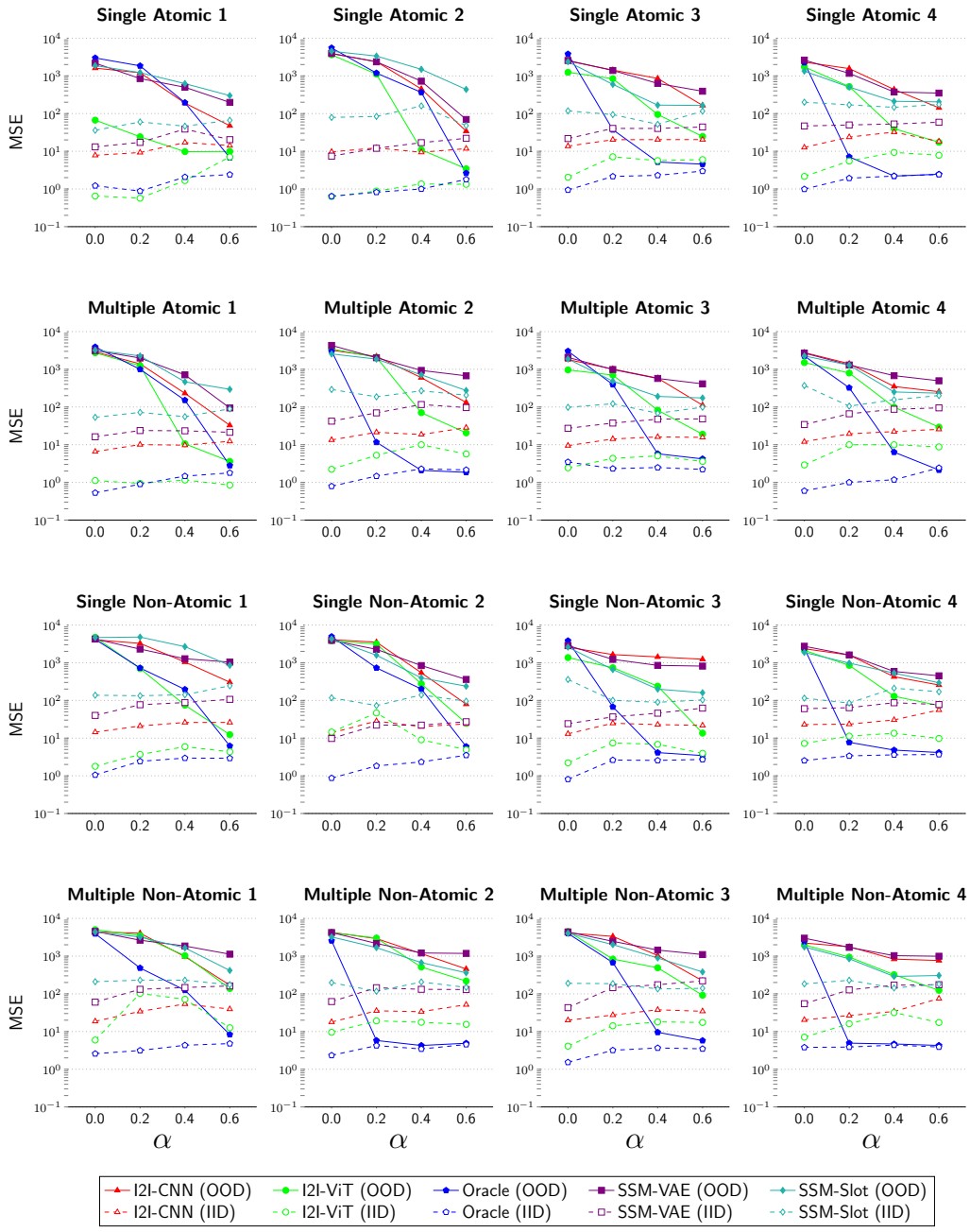

Figure 15: **MSE Performance on Analysis Tasks.**

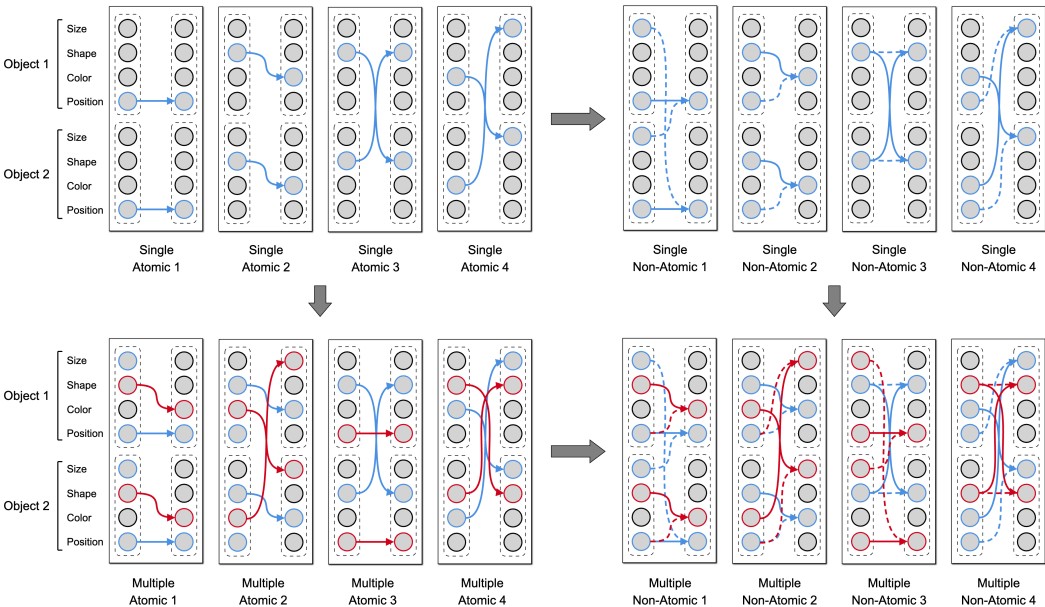

Figure 16: **Design of Analysis Set**. In this figure, we illustrate the 16 analysis rules using causal graphs, where nodes represent factors. The value of each target node is determined by the source nodes connected by incoming edges. If a node has just one incoming edge, the integer value of the source factor is directly assigned to the target factor, taking into account the cardinality of the target factor's vocabulary through modulo operation. When a node has two incoming edges, the integer values from both source factors are summed and then assigned to the target factor, subject to modulo based on the cardinality of the target factor's vocabulary. For example, in the *"Single Non-Atomic 4"* rule, two incoming arrows to the 'size' factor signify that its modified value is calculated by adding the quadrant/position of the same object to the color index of the other object, followed by a modulo operation with respect to the cardinality of object size vocabulary. All 16 rules depicted in this diagram can be understood in a similar manner.

Table 8: **LPIPS Performance of SVIB-dSprites.** We report the model performances on SVIB-dSprites for three levels of difficulty: Easy, Medium, and Hard, corresponding to $\alpha$ values of 0.6, 0.4, and 0.2, respectively.

|  | Models | Single Atomic | Multiple Atomic | Single Non-Atomic | Multiple Non-Atomic | Average |
|---|---|---|---|---|---|---|
| Easy | I2I-CNN | 0.0304 | 0.0209 | 0.1524 | 0.0536 | 0.0643 |
|  | I2I-ViT | 0.0036 | 0.0057 | 0.0050 | 0.0191 | 0.0084 |
|  | SSM-VAE | 0.0796 | 0.1203 | 0.1499 | 0.1638 | 0.1284 |
|  | SSM-Slot | 0.0274 | 0.0892 | 0.0228 | 0.0740 | 0.0534 |
|  | Oracle | 0.0006 | 0.0003 | 0.0005 | 0.0009 | 0.0006 |
| Medium | I2I-CNN | 0.1209 | 0.0844 | 0.1703 | 0.1249 | 0.1251 |
|  | I2I-ViT | 0.0172 | 0.0133 | 0.0535 | 0.0597 | 0.0359 |
|  | SSM-VAE | 0.1251 | 0.1591 | 0.1483 | 0.1735 | 0.1515 |
|  | SSM-Slot | 0.0206 | 0.1541 | 0.0280 | 0.1241 | 0.0817 |
|  | Oracle | 0.0007 | 0.0004 | 0.0006 | 0.0007 | 0.0006 |
| Hard | I2I-CNN | 0.2514 | 0.2780 | 0.2336 | 0.3082 | 0.2678 |
|  | I2I-ViT | 0.1700 | 0.2999 | 0.1361 | 0.3243 | 0.2326 |
|  | SSM-VAE | 0.2398 | 0.2748 | 0.1975 | 0.2519 | 0.2410 |
|  | SSM-Slot | 0.1022 | 0.2759 | 0.1324 | 0.2229 | 0.1834 |
|  | Oracle | 0.0069 | 0.0021 | 0.0156 | 0.0010 | 0.0064 |

Table 9: **LPIPS Performance of SVIB-CLEVR.** We report the model performances on SVIB-CLEVR for three levels of difficulty: Easy, Medium, and Hard, corresponding to $\alpha$ values of 0.6, 0.4, and 0.2, respectively.

|  | Models | Single Atomic | Multiple Atomic | Single Non-Atomic | Multiple Non-Atomic | Average |
|---|---|---|---|---|---|---|
| Easy | I2I-CNN | 0.0506 | 0.0380 | 0.0683 | 0.0802 | 0.0593 |
|  | I2I-ViT | 0.0487 | 0.0566 | 0.0468 | 0.0811 | 0.0583 |
|  | SSM-VAE | 0.0602 | 0.0768 | 0.0648 | 0.1034 | 0.0763 |
|  | SSM-Slot | 0.0288 | 0.0552 | 0.0839 | 0.0706 | 0.0596 |
|  | Oracle | 0.0101 | 0.0113 | 0.0110 | 0.0103 | 0.0107 |
| Medium | I2I-CNN | 0.0967 | 0.1238 | 0.1166 | 0.1692 | 0.1265 |
|  | I2I-ViT | 0.1313 | 0.0848 | 0.1425 | 0.1466 | 0.1263 |
|  | SSM-VAE | 0.0664 | 0.0962 | 0.0914 | 0.1303 | 0.0961 |
|  | SSM-Slot | 0.0605 | 0.0471 | 0.0517 | 0.0540 | 0.0533 |
|  | Oracle | 0.0116 | 0.0247 | 0.0084 | 0.0118 | 0.0141 |
| Hard | I2I-CNN | 0.1730 | 0.2502 | 0.2080 | 0.2410 | 0.2181 |
|  | I2I-ViT | 0.1899 | 0.2301 | 0.2005 | 0.2405 | 0.2153 |
|  | SSM-VAE | 0.1440 | 0.2604 | 0.1620 | 0.1871 | 0.1884 |
|  | SSM-Slot | 0.0973 | 0.1991 | 0.0630 | 0.0960 | 0.1139 |
|  | Oracle | 0.0124 | 0.0131 | 0.0113 | 0.0119 | 0.0122 |

Table 10: **LPIPS Performance of SVIB-CLEVRTex.** We report the model performances on SVIB-CLEVRTex for three levels of difficulty: Easy, Medium, and Hard, corresponding to $\alpha$ values of 0.6, 0.4, and 0.2, respectively.

| | Models | **Single Atomic** | **Multiple Atomic** | **Single Non-Atomic** | **Multiple Non-Atomic** | **Average** |
|---|---|---|---|---|---|---|
| Easy | I2I-CNN | 0.2803 | 0.2550 | 0.3642 | 0.2757 | 0.2938 |
| | I2I-ViT | 0.2293 | 0.2120 | 0.3381 | 0.2322 | 0.2529 |
| | SSM-VAE | 0.3504 | 0.3978 | 0.3832 | 0.3962 | 0.3819 |
| | SSM-Slot | 0.3044 | 0.3595 | 0.3442 | 0.3634 | 0.3429 |
| | Oracle | 0.1538 | 0.1268 | 0.1514 | 0.1334 | 0.1414 |
| Medium | I2I-CNN | 0.2751 | 0.2671 | 0.3678 | 0.2957 | 0.3014 |
| | I2I-ViT | 0.2277 | 0.2300 | 0.3354 | 0.2524 | 0.2614 |
| | SSM-VAE | 0.3452 | 0.3917 | 0.3817 | 0.3971 | 0.3789 |
| | SSM-Slot | 0.2906 | 0.3449 | 0.3578 | 0.3494 | 0.3357 |
| | Oracle | 0.1543 | 0.1306 | 0.1492 | 0.1434 | 0.1444 |
| Hard | I2I-CNN | 0.2819 | 0.3107 | 0.3651 | 0.3264 | 0.3210 |
| | I2I-ViT | 0.2625 | 0.3017 | 0.3145 | 0.3013 | 0.2950 |
| | SSM-VAE | 0.3440 | 0.3866 | 0.3758 | 0.3916 | 0.3745 |
| | SSM-Slot | 0.3106 | 0.3946 | 0.3546 | 0.3762 | 0.3590 |
| | Oracle | 0.1566 | 0.1221 | 0.1473 | 0.1356 | 0.1404 |

Table 11: **MSE Performance.** Numerical values of model performances on all 12 tasks: single atomic (S-A), single non-atomic (S-NA), multiple atomic (M-A), and multiple non-atomic (M-NA) tasks for each subset i.e., SVIB-dSprites, SVIB-CLEVR and SVIB-CLEVRTex.

| α | Models | SVIB-dSprites | | | | SVIB-CLEVR | | | | SVIB-CLEVRTex | | | |
|---|---|---|---|---|---|---|---|---|---|---|---|---|---|
| | | S-A | M-A | S-NA | M-NA | S-A | M-A | S-NA | M-NA | S-A | M-A | S-NA | M-NA |
| 0.0 | I2I-CNN | 2493.24 | 3201.78 | 2581.57 | 4273.86 | 884.86 | 1566.50 | 899.68 | 1520.91 | 647.37 | 752.06 | 988.72 | 945.42 |
| | I2I-ViT | 1242.31 | 3479.05 | 1367.78 | 4220.45 | 637.14 | 1642.70 | 634.05 | 1485.16 | 591.58 | 687.23 | 1020.89 | 932.65 |
| | SSM-VAE | 2562.39 | 4300.64 | 2862.57 | 4228.19 | 975.61 | 1546.94 | 959.04 | 1330.32 | 624.35 | 1048.46 | 713.09 | 1019.86 |
| | SSM-Slot | 2367.21 | 2557.13 | 2588.47 | 3232.08 | 897.26 | 1582.44 | 851.21 | 1187.62 | 643.18 | 1038.00 | 721.13 | 912.87 |
| | Oracle | 3798.89 | 3042.49 | 3822.39 | 2561.94 | 1366.79 | 1132.49 | 1116.75 | 851.99 | 817.70 | 983.48 | 944.21 | 936.89 |
| 0.2 | I2I-CNN | 1424.42 | 2193.89 | 1643.04 | 2922.01 | 402.27 | 1050.83 | 485.51 | 800.93 | 465.28 | 655.86 | 717.17 | 680.10 |
| | I2I-ViT | 843.12 | 2081.99 | 738.00 | 3015.15 | 438.48 | 974.72 | 461.85 | 808.84 | 495.31 | 614.44 | 738.57 | 666.23 |
| | SSM-VAE | 1389.60 | 2027.40 | 1227.31 | 2169.14 | 312.09 | 1100.34 | 349.47 | 542.07 | 540.06 | 871.26 | 657.42 | 819.19 |
| | SSM-Slot | 594.88 | 1862.77 | 659.48 | 1681.50 | 194.07 | 784.53 | 126.25 | 185.84 | 411.69 | 698.89 | 518.68 | 682.04 |
| | Oracle | 37.34 | 11.61 | 67.42 | 5.79 | 14.04 | 13.01 | 9.13 | 10.98 | 85.46 | 71.16 | 67.91 | 65.98 |
| 0.4 | I2I-CNN | 855.21 | 600.41 | 1424.66 | 1167.54 | 208.13 | 467.12 | 243.76 | 512.15 | 384.37 | 662.02 | 522.80 | 554.77 |
| | I2I-ViT | 94.72 | 70.60 | 238.69 | 514.59 | 299.26 | 327.33 | 315.14 | 417.57 | 322.02 | 562.63 | 428.36 | 446.58 |
| | SSM-VAE | 632.20 | 933.52 | 850.20 | 1213.25 | 118.25 | 300.71 | 181.64 | 313.25 | 536.78 | 894.49 | 655.99 | 829.35 |
| | SSM-Slot | 167.65 | 710.60 | 199.35 | 673.60 | 104.18 | 99.62 | 90.80 | 99.37 | 338.61 | 632.98 | 463.35 | 554.50 |
| | Oracle | 5.18 | 2.09 | 4.12 | 4.26 | 10.65 | 41.54 | 6.79 | 11.82 | 78.14 | 71.79 | 78.43 | 68.96 |
| 0.6 | I2I-CNN | 164.47 | 131.91 | 1237.73 | 456.57 | 104.72 | 84.50 | 125.46 | 186.00 | 383.99 | 615.90 | 391.05 | 418.67 |
| | I2I-ViT | 24.72 | 20.64 | 13.55 | 218.48 | 74.18 | 103.54 | 68.12 | 144.00 | 254.16 | 535.51 | 249.46 | 284.72 |
| | SSM-VAE | 391.55 | 671.98 | 813.27 | 1180.91 | 103.26 | 193.08 | 112.05 | 228.15 | 516.31 | 840.56 | 725.84 | 805.82 |
| | SSM-Slot | 164.16 | 274.67 | 159.92 | 361.76 | 45.31 | 91.23 | 172.18 | 121.40 | 319.21 | 536.25 | 427.24 | 511.08 |
| | Oracle | 4.54 | 1.85 | 3.44 | 4.89 | 9.86 | 11.51 | 9.15 | 9.76 | 76.99 | 74.22 | 71.38 | 64.50 |