# OpenReview forum: "Imagine the Unseen World: A Benchmark for Systematic Generalization in Visual World Models"
_NeurIPS.cc/2023/Track/Datasets_and_Benchmarks — NeurIPS 2023 Datasets and Benchmarks Poster_

### Official Review · Reviewer_zB71 · 2023-07-19
**A benchmark for systematic compositionality**

**Rating:** 6
**Confidence:** 3

**Strengths:**

I believe that this paper tackles an interesting problem for the research community. Development of reasoning agents is a commonly tackled problem related to several current problems (visual reasoning, action planning, or object-centric representations).

It is good to see authors providing a way to measure the ease of generalisation for the reasoning model through the parameter controlling a subset of seen in training combinations. I believe this to be a useful evaluation tool for assessing the models.

A selection of models for evaluation is good and I appreciate authors providing the oracle model as the upper bound for the performance.

**Additional Feedback:**

References:

[1] François Chollet. On the measure of intelligence. arXiv preprint arXiv:1911.01547, 2019.

[2] Rim Assouel, Pau Rodriguez, Perouz Taslakian, David Vazquez, and Yoshua Bengio. Object389 centric compositional imagination for visual abstract reasoning. In ICLR2022 Workshop on
390 the Elements of Reasoning: Objects, Structure and Causality, 2022.



**Clarity:**

The paper is clear and easy to follow. I would only suggest adding a table summarising the results.

**Correctness:**

Dataset is constructed in a sound way. It contains a large number of samples of varying difficulty. Suggested evaluation methods are appropriate for the benchmark.

**Documentation:**

A datasheet for dataset is provided along the submission. Authors mention that full benchmark will be released upon acceptance.

**Ethics:**

No ethical concerns.

**Limitations:**

I believe that societal impact of this work is discussed sufficiently.

However, I would like to see more discussion on the limitations of the benchmark and potential models. I think that the choice of the rules for the tasks is worth describing in the context of limitations, and as aforementioned putting it in the context of more general models would be valuable.

**Opportunities For Improvement:**

Firstly, the models are heavily dependant on the visual recognition performance. To this end, did authors consider rendering images in higher resolution? Did you have a chance to make a small ablation making sure that 128x128 is enough even for highly textured data?

Following on the above, I think it would be really interesting to introduce a model operating on disentangled representations into the evaluation. That would require learning a model generating images from such a representation as well, however, it could yield some interesting insights if dynamic action were to be performed in disentangled space. Additionally, with a recent huge growth of LLMs, it could also be a viable and interesting evaluation to include (e.g. with image input or with oracle input). It would be nice if authors could discuss such a potential direction of evaluation.

Further, I would like to ask authors how the core combinations were chosen. Specifically, when cardinality of concepts is not equal within the dataset (e.g. we have a set of 8 shapes and 3 sizes). Does that affect generalisability?

It would also be nice to see more explanation of certain design choices, such as how the tasks are designed. In the benchmark we have 4 subsets per dataset corresponding to different rules (e.g. new shape dependant on all shapes in the scene). How those rules were created?

Following the design of rules, it would be good to see some comment, possibly an experiment on potential of the use of the data to train more general model, e.g. in a few-shot scenario. This could be a very valuable insight into more general models, similar to what ARC [1], and Sort-of-ARC [2] tried to test. Further, appendix D.1.2 mentions 16 analysis tasks, 4 per each of 4 atomic rules, followed by mention that dSprites use only 1 each. How many do CLEVR-based datasets use (from datasheet it seems 4 as well)? Could you elaborate on those 16 tasks?

In the oracle baseline, how is decoder trained? I wonder if there is a significant impact of image generator onto the whole system performance.

**Relation To Prior Work:**

Related works are extensively cited and compared to.

**Summary And Contributions:**

This work proposes a new benchmark focused on evaluating systematic compositionality. The task of the system is to understand the composition of the scene based on the set of observations and predict a novel, previously unseen composition of previously seen concepts. To this end authors propose a set of benchmarks constituting short two-frames episodes based on some well-recognised datasets, in which the system has to imagine target frame based on the source one. Along the benchmark, authors introduced a set of experiments to evaluate state-of-the-art models performance in this new task.

---

> ### Author Response · Authors · 2023-08-21
> **Response to Reviewer zB71 [1/2]**
>
> We sincerely thank you for your positive recommendation and very helpful questions and comments!
>
> > The models are heavily dependent on the visual recognition performance. Perform ablation to show that 128x128 is enough for highly textured data.
> >
>
> Thank you for bringing up this question. We have found that 128x128 image resolution is not an issue for highly textured data. To support this claim, we conducted an experiment in which we trained a simple 4-layer CNN encoder to take CLEVR-Tex images as input and predict GT labels. We found that this model was able to predict the factors with high accuracy (88-100% accuracy). A more powerful encoder is expected to perform even better. In our revised supplementary, we have added the results of this experiment in Table 4.
>
> > It would be really interesting to introduce a model operating on disentangled representations into the evaluation. That would require learning a model generating images from such a representation as well, however, it could yield some interesting insights if dynamic action were to be performed in disentangled space.
> >
>
> We believe that two of our baselines already cover the suggested experiment. Please let us know if the following description does not answer your question.
>
> - Oracle Model: In this model, we construct a representation of the input scene by taking each ground truth factor of the scene, representing it as a vector, and then stacking these vectors together. This produces a perfectly disentangled representation with true factor values.
> - SSM-VAE: In this model, we first pre-train a VAE on the images of the task episodes. VAE naturally produces a disentangled representation due to the KL term in its training objective. Therefore, this baseline also operates on a disentangled representation, although it may not be as accurate as the Oracle model.
>
> > Additionally, with a recent huge growth of LLMs, it could also be a viable and interesting evaluation to include (e.g. with image input or with oracle input). It would be nice if authors could discuss such a potential direction of evaluation.
> >
>
> The reviewer raises an interesting point that LLMs could provide the priors necessary to systematically generalize and perform well on our tasks. We appreciate your pointing out this interesting aspect. In the revision, we have discussed this potential as suggested. Indeed, our benchmark is designed to facilitate these kinds of investigations.
>
> > Further, I would like to ask authors how the core combinations were chosen. Specifically, when cardinality of concepts is not equal within the dataset (e.g. we have a set of 8 shapes and 3 sizes). Does that affect the generalization aspect?
> >
>
> When the cardinality of factors is not equal, the primitives for the factor with smaller cardinality appear more than once. Taking your example of 8 shapes and 3 sizes, we can illustrate the core combinations as follows:
>
> |  | Shape 1 | Shape 2 | Shape 3 | Shape 4 | Shape 5 | Shape 6 | Shape 7 | Shape 8 |
> | --- | --- | --- | --- | --- | --- | --- | --- | --- |
> | Size 1 | Core |  |  | Core |  |  | Core |  |
> | Size 2 |  | Core |  |  | Core |  |  | Core |
> | Size 3 |  |  | Core |  |  | Core |  |  |
>
> where columns correspond to the 8 shape primitives and rows correspond to the 3 size primitives. In this way, our training set still exposes all primitives at least once and does not violate systematicity. We have now clarified this in Appendix D.2 in our revision.
>
> > It would also be nice to see more explanation of certain design choices, such as how the tasks are designed. In the benchmark we have 4 subsets per dataset corresponding to different rules (e.g. new shape dependent on all shapes in the scene). How those rules were created?
> >
>
> Thank you for this question. The process of choosing the rules is indeed non-trivial as the number of possible rules is large (~65K possible causal graphs for dSprites). We have explained our rule selection process in Appendix D.1.2. To summarize the key points, our rule selection takes the following points into consideration:
>
> 1. Have one rule for each complexity level: single atomic, single non-atomic, multiple atomic, and multiple non-atomic.
> 2. Balance difficulty and solvability. That is, while the rules should make various baselines struggle, they should also be solvable by the Oracle.

---

> > ### Author Response · Authors · 2023-08-21
> > **Response to Reviewer zB71 [2/2]**
> >
> > > Following the design of rules, it would be good to see some comment, possibly an experiment on **potential of the use of the data to train more general model**, e.g. in a few-shot scenario. This could be a very valuable insight into more general models, similar to what ARC [1], and Sort-of-ARC [2] tried to test.
> > >
> > - Our dataset targets zero-shot learning, instead of few-shot learning. That is, it asks whether the model can handle compositionally novel inputs without any further training at test time. This is the main difference between our dataset and ARC or Sort-of-ARC. The ARC or Sort-of-ARC dataset is not designed to test this zero-shot ability. To do this, our dataset assumes global world dynamics which is not changed at test time.
> > - Nevertheless, we agree that it would be a great future extension of our benchmark to study compositional systematicity in both aspects, i.e., in terms of the visual inputs and dynamics. In our revised version, we have noted this aspect in Section 7.1.
> >
> > [1] Chollet, François. “On the Measure of Intelligence.”
> >
> > [2] Assouel, Rim et al. “Object-centric Compositional Imagination for Visual Abstract Reasoning”
> >
> > > Further, appendix D.1.2 mentions 16 analysis tasks, 4 per each of 4 atomic rules, followed by mention that dSprites use only 1 each. How many do CLEVR-based datasets use (from datasheet it seems 4 as well)? Could you elaborate on the 16 analysis tasks?
> > >
> >
> > Yes, you are correct. All 3 environments use the same 4 rules as mentioned in the datasheet, resulting in a total of 12 tasks.
> >
> > For performing evaluation on 16 analysis rules, we conducted experiments only on the dSprites environment due to the computational cost of training 5 baselines across 16 tasks with 4 training splits per task, totaling 320 training jobs. Based on the dSprites analysis results, we selected 4 rules and then used the same 4 rules across all 3 environments to construct our final benchmark.
> >
> > Thank you for asking about an elaboration on the 16 analysis tasks. In Figure 17, we diagrammatically describe each of the 16 analysis rules. We agree that a better description was needed. Therefore, in our revised version, we have expanded the caption of the diagram to better guide the reader on how to interpret the diagram.
> >
> > > In the oracle baseline, how is decoder trained? I wonder if there is a significant impact of image generator onto the whole system performance.
> > >
> >
> > That is a great question. To train the decoder, we took care to use an identical decoder architecture (i.e., a transformer) and an identical training objective in all our baselines, including Oracle. As such, the sole contributing factor to the superior performance of Oracle is the factored GT representation and not the decoder.
> >
> > > However, I would like to see more discussion on the **limitations of the benchmark and potential models**. I think that the choice of the rules for the tasks is worth describing in the context of limitations, and as aforementioned putting it in the context of more general models would be valuable.
> > >
> >
> > Thank you for pointing out this. We have added a limitation section in the paper to describe these (and other) limitations of our benchmark (Section 7.1).
> >
> > Regarding dynamics, we list some limitations below. Note that these choices were deliberately made to keep focus on the systematicity of the visual input rather than dynamics.
> >
> > - Our dynamics are simple e.g., swapping shape or updating size as a function of its color. Such simplicity is introduced deliberately to make analysis easy. However, more complex e.g., dynamics involving multiple steps or dynamics with hierarchical temporal structure, would be natural avenues for future extension.
> > - As mentioned earlier, our dynamics are fixed within a task. In this respect, a natural extension of our benchmark would be to have varying dynamics that need to be inferred in a few-shot manner at test time.
> >
> > > I would only suggest adding a table summarising the results.
> > >
> >
> > Thank you, we will add such a table in our final version.

---

> > > ### Comment · Reviewer_zB71 · 2023-08-29
> > > **Reviewer resp[onse**
> > >
> > > Thank you for addressing the concerns I raised in the review. I appreciate the explanations and clarifications provided as well as the experiment addressing the raised question.

---

> > > > ### Author Response · Authors · 2023-08-30
> > > > **Follow-Up Response to Reviewer zB71**
> > > >
> > > > Thank you!

---

### Official Review · Reviewer_ikkV · 2023-07-20
**A helpful framework to design and test abstractive representation of visual features.**

**Rating:** 9
**Confidence:** 4
**Clarity:** This paper is well written.

**Strengths:**

Strengths:
The task design to probe the question of visual imagination based on abstract visual cues is direct and creative.
The images synthesized here used object centric scene generation, essentially from 3 rendering environment covering 2D, 3D and 3D with texture.
All images are multi-object, and different sets of visual factors are used to modify intra-object visual primitives, which has to be achieved based on abstract concept representation of those primitives.
One important advantage is the out-of-distribution design of test set primitives, where the combination of visual primitives do not overlap with those in the training.
The question, benchmark design and example baseline models can make a clear contribution to the research field.

**Additional Feedback:**

None.

**Correctness:**

The dataset is carefully designed and constructed in a sound way. The metric to evaluate performance not only utilized MSE, which reflects low level errors, but also higher level metric LPIPS, which has been used to evaluate human perception. The approach is appropriate.

**Documentation:**

The documentation and code repo is clearly organized. The paper and documentation in the benchmark is good for reproducibility. It would be helpful to include tutorial for image rendering from different 2D/3D models in the code (if not there yet).

**Ethics:**

No.

**Limitations:**

The potential negative societal impact is adequately discussed.

**Opportunities For Improvement:**

Is it necessary to design 'swapping' case between the two objects, meaning the rule is applied symmetrically to the two objects. What is the advantage of this symmetry of rule change? Is it possible to apply different rules to the two object in the same scene? How about more objects?

How to improve the ability to form or encode 'mental model' is still a bit unclear given the evaluation of the benchmark. This is a question beyond the current scope, but some intuition would be helpful. For example, the baseline models have different designs. The limitation of each model to form a good relational representation might be different.

Summary about previous work can be extended, especially given the fast evolving field of visual prompt (see comments below).


**Relation To Prior Work:**

The idea to utilize conceptual or abstractive rules to generate new images is not new. Generative models have explored this direction as well. It would be helpful to summarize work in the domain of image rendering/synthesis based on semantic prompt and especially visual prompt, and compare that strategy (and its potential 'world model' implication) with current study.


**Summary And Contributions:**

The ability to form a mental model composed of abstract concepts is a fundamental intellectual ability of human beings but remains challenging for machine learning. While the tokenized or compositional structure is more natural in language processing, it is less explored in visual processing. This paper tackled the problem heads-on by designing a systematic 'visual imagination' task, which requires the model to generate an 'imagined' image for a given input image based on composition of visual factors, like color, shape, size, etc. They consider this one-step image-to-image transformation task using abstract visual factors as a reflection of utilizing the underlying latent world representation, therefore a direct test of the visual "mental model".

By forming a benchmark of tasks in this domain, the paper tested different models against this benchmark to evaluate the current status of 'visual imagination' in deep learning models. This paper have tested 5 different baseline models in their benchmark. The image-to-image model has three realizations: 1. CNN based encoder-decoder; 2. Transformer (ViT) based encoder-decoder; 3. Transformer based but taking vectors as input instead of images (Oracle). The state-space model, which first encode images to a latent space and generate a predicted image y with a decoder, has two realizations: 1. SSM-VAE, which is VAE based model; and 2. SSM-SAVi, which is object-centric representation based model. Among the 5 baseline models, Oracle showed best tolerance to variations between input and target images.

---

> ### Author Response · Authors · 2023-08-21
> **Response to Reviewer ikkV**
>
> We are delighted by your positive recommendation and insightful suggestions!
>
> > **Why apply the same rule to the two objects? Is it possible to apply different rules to the two object? How about more objects?**
> >
>
> This is a great suggestion. The current design, based on symmetry, draws inspiration from universal laws of physics that apply equally to all physical objects. This choice also lends conceptual simplicity to the dataset. However, we agree that the non-symmetric case is also interesting. We believe though that a deeper understanding of the current dataset will better equip us to approach cases with more complex and general dynamics. Thank you again for the great suggestion! If we make the next version of the benchmark in the future, the suggestions should be the features of the highest priority.
>
> > **How to improve the ability to form or encode ‘mental model’? (A question beyond the current scope, but some intuition would be helpful.)**
> >
> - Thank you for suggesting this. This is indeed the key question we are also considering for future work. The general lesson from the current paper is that (1) we should focus on improving the feature factorization because perfect factorization (GT) performed the best, or (2) from the result that ViT also performs reasonably well, it seems worth investigating some encoder designs built upon ViT but with a proper level of inductive bias that encourages systematic generalization.
> - In fact, in the current version, we have provided some pointers for improving the models on our benchmark in Section 6.3 "Summary of Experiment Results". In our revision, we have highlighted this more clearly.
>
> > **Summary about previous work can be extended, especially given the fast-evolving field of visual prompts (see comments below).**
> >
>
> Thank you for this suggestion. Following the suggestion, we have discussed more works in the revision, including the works on image generation based on semantic prompts and visual prompts, and their potential implication as a 'world model'.  To name a few examples, we found the following papers interesting [1, 2, 3]. Also, we would deeply appreciate it if you could suggest any specific papers that come to mind.
>
> [1] Wang, Xinlong et al. “Images Speak in Images: A Generalist Painter for In-Context Visual Learning.”
>
> [2] Bar, Amir et al. “Visual Prompting via Image Inpainting.”
>
> [3] Hosseini, Arian et al. “On the Compositional Generalization Gap of In-Context Learning.” *BlackboxNLP Workshop on Analyzing and Interpreting Neural Networks for NLP* (2022).
>
> > **It would be helpful to include tutorial for image rendering from different 2D/3D models in the code (if not there yet).**
> >
>
> Thank you, this is a great point. We will include a tutorial in the code.

---

### Official Review · Reviewer_A3fU · 2023-07-23
**Interesting dataset on compositionality**

**Rating:** 6
**Confidence:** 3
**Clarity:** The paper is well-written and is easy…

**Strengths:**

This paper addresses a fundamental issue in visual reasoning, namely systematic compositionality. The design of the tasks directly targets this issue. The design is simple and clever.

The baseline models are also well-designed, and the experimental results indicate that existing model architectures are inadequate for such tasks. The paper has the potential to inspire future research on developing models that can handle compositionality.



**Additional Feedback:**

It can be desirable to have more discussions on how to improve the baseline models to deal with compositionality.

**Correctness:**

The dataset is constructed in a sound day, and the evaluation methods and experiment design are performed correctly. The claims made look correct to me.

**Documentation:**

The paper provides sufficient detail on data generation. There is sufficient detail to support reproducibility.

**Ethics:**

No ethical concern.

**Limitations:**

The authors may want to address the limitations of their dataset in the main text extensively. I do not see any potential negative societal impact.

**Opportunities For Improvement:**

While the design of the dataset is very sensible, the dataset is rather artificial and simplistic. The images contain very simple primitives with simple attributes, and the dynamics follows simple rules. It is unclear how relevant such a dataset is to real image or video data or realistic visual reasoning tasks. Rendering more realistic images with complex primitives and dynamics may lead to improved dataset.

**Relation To Prior Work:**

The related papers are adequately discussed in the main text. The difference from previous contributions is clearly discussed.

**Summary And Contributions:**

This paper proposes to use synthetic image pairs to test systematic compositionality. The image primitives are compositions of various attributes, such as color, shape, and position. The dynamics or transformation is based on a rule of varying complexity. The task is to predict the transformed image given the original image. The task is tested on baseline models including image-to-image models and state-space models. The baseline models fail when the training compositions and testing compositions are different.

---

> ### Author Response · Authors · 2023-08-21
> **Response to Reviewer A3fU**
>
> We sincerely thank you for your positive assessment of our benchmark!
>
> > While the dataset is very sensible, it is rather artificial and simple. More realistic images can improve the dataset.
> >
>
> We appreciate that you found our dataset sensible! We agree that the proposed dataset might be viewed as artificial and not realistic.
>
> 1. However, it is crucial to note that current methods are all failing on this dataset.
> 2. Additionally, there is a trade-off between complexity (or realism) and ease of analysis. Being synthetic, we have access to the ground truth which makes analysis easy. That being said, our benchmark does not need to be the only task for studying world model compositionality. The community can try to proceed towards more realistic tasks after understanding the problem more deeply within our dataset. Yet, our synthetic benchmark can play a role similar to how the MNIST dataset has played an important role.
> 3. In fact, to our knowledge, our CLEVRTex tasks are much more visually complex than any existing generalization benchmarks in the visual domain [1, 2, 3].
>
> [1] Chollet, François. “On the Measure of Intelligence.”
>
> [2] Assouel, Rim. “Object-centric Compositional Imagination for Visual Abstract Reasoning”
>
> [3] Webb, Taylor W. et al. “Learning Representations that Support Extrapolation.”
>
> > Discuss Limitations of the Dataset
> >
>
> While we mention some limitations (e.g., in line 318), we agree that it is somewhat too brief and would be better to have a more detailed discussion of limitations, e.g., in a separate section. We have added a more in-depth discussion in the revision (Section 7.1).
>
> > Additional feedback: Desirable to have more discussion on how to improve the baselines
> >
>
> Thank you for suggesting this. We have revised Section 6.3 to better highlight the pointers for improving the baselines. From the fact that perfect factorization (GT) is the best, we can pursue a better factorization method. And, from the fact that ViT works reasonably, one could also pursue extensions of ViT architecture that can better encourage compositional generalization.

---

> > ### Comment · Reviewer_A3fU · 2023-08-22
> > **feedback**
> >
> > Thanks for the rebuttal, which has addressed my concerns.

---

> > > ### Author Response · Authors · 2023-08-30
> > > **Follow-Up Response to Reviewer A3fU**
> > >
> > > We are glad to hear this!

---

### Official Review · Reviewer_LyM3 · 2023-07-24
**Unclear contributions**

**Rating:** 6
**Confidence:** 3

**Strengths:**

The paper tackles an interesting problem of generalization from previous observations and being able to predict the next state of the world. This is a capability which human has and is still comparatively better to artificial systems.

**Additional Feedback:**

See my many questions above.

**Clarity:**

The paper is not well written, information is presented in wrong places, too many sections, too broad and too little specific in many areas.

Specifically:
- Introduction:
  - what task exactly you want to solve? General compositionality, dynamic compositionality? Working with language, only vision or both? Don't describe all options, but focus on the points related clearly to your contributions
  - how can systematic compositionality originate from philosophy? You mean the problem is described there?
   - "which simplifies addressing compositional systematicity" - what do you mean by this?
   - Why is VQA indirect?
   - What is the second group of works about?
   - Why is the 3rd group ignoring visual perceptual systematic generalization? Why you mention it and what does it mean?
   - how does your work relate to next frame video frame prediction (see Kitti benchmark), predictive coding etc?
   - many citations, select only the most important ones
   - ARC exhibit these limitations - which ones?

- Contributions in Introduction
  - it should be clear that the benchmark is a collection of dSprites, Clever and say what it gives on top on just evaluating the models on these datasets - dynamics? Specifically selected tasks?

- Fig.1 states it is overview of SVIB, but it is not clear to me from it what SVIB actually is
  - is the swap the only dynamic rule introduced in SVIB?

- Section 2
  - it is not clear which exact dynamic rules are introduced?
  - sec.2.1 should be clear from the beginning of sec.2

- sec.3 - metrics should be better described - lpips metric - not described at all
  - why/where it was shown that MSE overly focus on low-level errors - which errors?

- Sec.4 - would be good to evaluate some standard models, it is not clear to me what exactly is the design of your baselines. If these are just standard models, only cute the models, if you extended them, state clearly why extension was needed and how it was done

- Sec 5 - shortcut OOD not introduced
  - which alpha parameter? It just comes out of nothing

- Sec.6 - effect of alpha - which alpha? This should be described and introduced before

**Correctness:**

I have quite strong doubts about how this benchmark is constructed. Also evaluation metrics are not clear at all. However, it is difficult to evaluate thanks to the very unclear presentation.

**Documentation:**

Authors release only sample of the dataset, but do not release code to conduct experiments available and state it will be made available after acceptance of the paper. There is also no documentation how to conduct your own experiments.

**Limitations:**

I didn't find the limitations of the work described in the paper.

**Opportunities For Improvement:**

I was really confused while reading the paper:
- the introduction is very broad and doesn't come to the point. (See clarity section below)
- contributions are not clear
- in section 2.1 I finally found that the proposed benchmark is basically a collection of dSprites, Clever and Clevertex tasks- this should be clear from the introduction and should be the first thing to say in the Section 2 (way more important than number of episodes)
- is the added value in adding dynamics? If so, this should be clearly stated in Introduction. How is the dynamics tackling the compositionality problem - are the rules composed in a sound way?
- I don't see how should as a human be able to know that objects should swap and in parallel change color and size? How this relates to dynamic behavior of cat/dog etc as described in introduction?
- It is not clear to me if baselines are standard models or their adaptation - in experiments it is stated that they compare 5 baselines, these should be clearly separated and described in Sec.4 - maybe as bullet points or subsections, in sec.4, I don't even see Oracle method mentioned
- related work - I would give more focus on the existing benchmark datasets rather than on the methods as this works introduces a "novel" benchmark
- metrics - it is crucial for comparing the methods to have a good way to evaluate the generalization and prediction capabilities. The used metrics are not described well and on top I think that these are not well enough capturing the predictive and generalization capabilities of the models
- you state you have multiple objects - swap rule would be applied how in the case of multiple objects? Randomly? Based on some rule?

My main concern is about how the benchmark actually models the dynamics and generalization of the rules between various scene/ objects. Is there any meaning behind the modeled dynamics? I think authors should way more focus on clearly and specifically describing their contributions in this area.

**Relation To Prior Work:**

I don't see clever, dSprites, Sprites datasets mentioned in RW, although this benchmark builds on top of these...the relation to other similar datasets should be clearly stated

**Summary And Contributions:**

The paper tackles the problem of visual imagination of the next scene based on the previous observations. Compositionality of the scenes should be explored by this benchmark.

---

> ### Author Response · Authors · 2023-08-21
> **Response to Reviewer LyM3 [1/4]**
>
> ### Introduction
>
> > What task exactly you want to solve? General compositionality or dynamic compositionality? Working with language, only vision or both? Don't describe all options, but focus on the points related clearly to your contributions.
> >
>
> We've clearly stated our task in both the abstract and introduction.
>
> - **Line 63-69:** *“In this paper, we introduce a new benchmark called the Systematic Visual Imagination Benchmark (SVIB), the first benchmark to rigorously address the compositional visual imagination problem. The SVIB poses the task as a minimal world modeling problem: one-step image-to-image generation. The objective is to generate the subsequent scene image from the current one. The underlying world dynamics operate as a relational function of object-centric factors. The benchmark presents a subset of possible combinations of the visual primitives during training, and during testing, exclusively presents compositionally novel source images to assess a model’s systematic imagination ability.”*
>
> The reviewer has perhaps misunderstood our intention to provide a thorough background before stating our task (Lines 45-62). However, providing such background should be viewed as a strength, not a shortcoming, as it enables us to appropriately position our contributions.
>
> > How can systematic compositionality originate from philosophy? Do you mean that the problem is described there?
> >
>
> We mean that the **notion or idea** of systematicity and compositionality were first discussed in the domain of philosophy, as evidenced by the reference we have cited in our paper.
>
> > *"which simplifies addressing compositional systematicity"* - What do you mean by this?
> >
>
> We are contrasting the difficulty of approaching the problem of systematic generalization in the language domain versus in the vision domain. We argue that in the language domain, the problem is more approachable.
>
> This is because the building blocks for language, i.e., words, are predefined via a vocabulary [1]. Every possible sentence can be constructed via combinations of words within this vocabulary. However, in the visual domain, there is no clear-cut vocabulary of building blocks that can be reused in systematically novel scenes. Furthermore, the process of how a visual observation is produced from the underlying building blocks is much more complex than it is for language.
>
> [1] Lake, Brenden M. and Marco Baroni. “Generalization without Systematicity: On the Compositional Skills of Sequence-to-Sequence Recurrent Networks.” *International Conference on Machine Learning* (2017).
>
> > Why is VQA indirect?
> >
>
> VQA or Visual Question Answering tasks provide text alongside images. Such text labels can provide help (direct or indirect) about what the composable building blocks in those images are. Therefore, VQA tasks do not tackle the issue of systematic generalization in the visual domain in a direct manner.
>
> > What is the second group of works about?
> >
>
> The second group of works is those that pursue disentangled representation learning as a subgoal for achieving systematic generalization in neural networks. As such, this approach is also not direct.
>
> > Why is the third group ignoring visual perceptual systematic generalization? Why do you mention it and what does it mean? Which limitations does ARC exhibit?
> >
>
> We mention the third group of works because it contains two benchmarks closest to ours: ARC and Sort-of-ARC. Like ours, both benchmarks involve mapping unstructured input to an unstructured target. However, both ignore the problem of visual perceptual systematic generalization for the following reasons.
>
> 1. **ARC Benchmark:** This benchmark is not ideal for studying systematic generalization as the test instances may have primitives not encountered during training.
> 2. **Sort-of-ARC:** While this benchmark ensures a systematic train/test split, it focuses on the systematicity of dynamics rather than the systematicity of visual concepts e.g., color or shape.

---

> > ### Author Response · Authors · 2023-08-21
> > **Response to Reviewer LyM3 [2/4]**
> >
> > ### Body
> >
> > > In section 2.1, I finally found that the proposed benchmark is basically a collection of dSprites, CLEVR, and CLEVRTex tasks. This should be clear from the introduction and should be the first thing to say in Section 2.
> > >
> >
> > This is not true. To say that our work simply evaluates models on existing datasets completely misses the point. Our benchmark is not simply a repackaging of original dSprites/CLEVR/CLEVRTex. Our benchmark consists of entirely new tasks. There is no existing dataset (or their specific subsets) that can be used for the tasks presented in our paper.
> >
> > Specifically, the original dSprites/CLEVR/CLEVRTex datasets do not offer image-to-image prediction tasks with a train/test split designed to evaluate systematic generalization.
> >
> > Although we built upon the codebases of dSprites/CLEVR/CLEVRTex, creating our benchmark took months of intensive work and refinement involving:
> >
> > - Identifying what is unexplored in visual systematic generalization research.
> > - Designing the primitives (e.g., colors, shapes), the rules (e.g., shape-swap), and the tasks.
> > - Developing the code base and rendering scenes.
> > - Extensively experimenting to create tasks that present a difficult challenge to the current state-of-the-art while also remaining solvable.
> >
> > We think the reviewer has misunderstood our contribution and we hope that our response will reform their views.
> >
> > > Is the added value in adding dynamics? If so, this should be clearly stated in Introduction.
> > >
> >
> > It is true that our benchmark involves dynamics while original dSprites/CLEVR/CLEVRTex datasets are about static scenes. However, it is not just the dynamic modeling that is important, but **systematically generalizing to novel visual inputs**. To our knowledge, no benchmark focuses on this aspect. As discussed in our related work, there are many video benchmarks for dynamics modeling, however, they all focus on the IID setting, unlike ours.
> >
> > > How is the dynamics tackling the compositionality problem - are the rules composed in a sound way?
> > >
> >
> > Our dynamics or rules are based on the simplest functional forms. For instance, the Shape-Swap can be executed using a simple assignment operator. The shape of object 1 is assigned to object 2 and the shape of object 2 is assigned to object 1. We think, as a systematic generalization task, it is reasonable to expect that a model that can swap the shapes of a “red sphere” and “blue cube” should be able to swap the shapes of a “blue sphere” and “red cube” when asked at test time.
> >
> > Despite this simplicity of the dynamics, our experiments show that a variety of models in use today struggle to systematically generalize.
> >
> > > I don't see how should a human be able to know that objects should swap and in parallel change color and size.
> > >
> >
> > It is a subjective question whether a human should be able to solve a task or not. However, as far as neural networks are concerned, our experiments show that our benchmark does not pose an unsurmountable challenge. For this, we performed an evaluation of an Oracle model—a model which can access ground-truth (GT) factors of the scene. We find that the Oracle can learn the rule well and systematically generalize in applying it to systematically novel scenes as early as $\alpha=0.2$.
> >
> > > How this relates to the dynamic behavior of the cat and dog described in the introduction?
> > >
> >
> > The concepts “small”, “cat”, “big” and “dog” in this scenario are analogous to various factor primitives in our benchmark e.g., color, shape, size, etc. Like the cat and dog scenario, in our benchmark, we expect a model trained only on a subset of possible factor combinations to effectively handle unseen combinations of familiar factors during testing.
> >
> > > From Figure 1, is the swap the only dynamic rule introduced in SVIB? In Section 2, it is not clear which exact dynamic rules are introduced.
> > >
> >
> > No, the shape swap is not the only rule introduced in SVIB. There are several tasks, each based on a specific rule. This is clear from Lines 142-145:
> >
> > “*In total, our benchmark provides 12 tasks containing 4 tasks per each rendering environment: dSprites, CLEVR, and CLEVRTex. Within each environment, the 4 tasks correspond to the 4 rule categories: single atomic, single non-atomic, multiple atomic, and multiple non-atomic. For formal definitions of the 4 rules, see Appendix D.1.1.”*
> >
> > We have also left a pointer to the relevant section of the appendix where further details are provided. As mentioned in the paper, we also provide a preview of our dataset at the link: https://systematic-visual-imagination.github.io/.
> >
> > > How would the swap rule be applied in the case of multiple objects?
> > >
> >
> > Our scenes are currently confined to 2 objects. As ours is the first benchmark for compositional world modeling, we have adopted a design that is as simple as possible while posing a challenge for the current models. Having more objects and designing appropriate dynamics/rules for them is a future direction.

---

> > > ### Author Response · Authors · 2023-08-21
> > > **Response to Reviewer LyM3 [3/4]**
> > >
> > > ### Metrics
> > >
> > > > Metrics should be better described - LPIPS metric.
> > > >
> > >
> > > Since LPIPS is a standard metric in the vision literature for measuring perceptual similarity, it is common to only cite the paper and omit a detailed description. Broadly, LPIPS works by taking two images as input, extracting their features via pre-trained large-scale image classification models (AlexNet in our case), and computing the distance between the extracted features. It has been shown by the original authors that LPIPS is an error metric that better corresponds to human judgment. We use the official implementation of LPIPS [1, 2].
> > >
> > > [1] https://github.com/richzhang/PerceptualSimilarity/
> > >
> > > [2] Zhang, Richard et al. “The Unreasonable Effectiveness of Deep Features as a Perceptual Metric.” *2018 IEEE/CVF Conference on Computer Vision and Pattern Recognition* (2018): 586-595.
> > >
> > > > Why and where was it shown that MSE overly focuses on low-level errors - which errors?
> > > >
> > >
> > >  By low-level errors, we mean small deviations in pixel values e.g., small blurriness or small deviations in pose, that don’t affect the overall human perception of the predicted image. MSE simply computes pixel-wise L2 distance between the predicted and the target images. As such it harshly penalizes such low-level errors.
> > >
> > > > Metrics: It is crucial for comparing the methods to have a good way to evaluate the generalization and prediction capabilities. The used metrics are not described well and on top, I think that these are not well enough to capture the predictive and generalization capabilities of the models.
> > > >
> > >
> > > We think the description of our metrics is clear since another reviewer (iKKV) has summarized our metrics accurately. Regarding the metrics themselves, it would help us if you can specify in what way our metrics fall short. We'd be happy to clarify.
> > >
> > > ### Baselines
> > >
> > > > In Section 4, it would be good to evaluate some standard models. It is not clear to me what exactly is the design of your baselines. If these are just standard models, only cite the models, if you extended them, state clearly why the extension was needed and how it was done.
> > > >
> > >
> > > Our baselines are standard neural network layers widely used in the community for scene representation: CNN, Vision Transformer (ViT), Slot Attention, and VAE. We have used their standard implementations. Moreover, we have already cited the papers that have introduced these models. We have also already described the hyperparameters of these baselines in Table 5 in the appendix.
> > >
> > > It would help us if you can let us know what exactly confused you about the design. We would be happy to clarify.  Furthermore, we are committed to releasing the complete codebase including the baseline implementations upon acceptance as we already mention in our reproducibility statement.
> > >
> > > > Describe baselines as bullet points or subsections.
> > > >
> > >
> > > Thank you for this suggestion. We will consider it for our final revision.
> > >
> > > > In Section 4, I don't even see the Oracle method mentioned.
> > > >
> > >
> > > This is not true. The Oracle is clearly mentioned in Section 4.
> > >
> > > *“Finally, the Oracle baseline is a variant of this (Image-to-Image) category where instead of encoding the input image to obtain the representation $e$, we directly construct and provide a collection of vectors as input, where each vector represents a ground truth factor of the scene. More details are provided in Appendix E.3.”*
> > >
> > > We have also left a pointer to the relevant section of the appendix where more details are provided.
> > >
> > > ### Acronyms and Notations
> > >
> > > > Acronym OOD is not introduced.
> > > >
> > >
> > > OOD stands for Out-of-Distribution. We have added this in our revised version.
> > >
> > > > Which alpha parameter? It comes out of nothing.
> > > >
> > >
> > > This is not true. The $\alpha$ parameter is described clearly in several places within Section 2 as follows:
> > >
> > > - Line 115: “*To present mutually exclusive primitive combinations in training and testing, we set aside 20% of the non-core combinations for testing. After setting aside the core combinations and the testing combinations, we then **select a fraction $\alpha$ of the remaining combinations and add the core combinations to obtain the training combinations.***”
> > > - The caption of Figure 2: “*In training episodes, we present all shape and color primitives individually, however, we do not show all possible combinations by defining a **parameter $\alpha$ that controls the fraction of the combinations presented***".
> > > - Line 145: “*For each task, **we provide 4 training splits corresponding to distinct $\alpha$ values** 0.0, 0.2, 0.4, and 0.6, and a common testing split that is systematically OOD with respect to all 4 training splits.*”

---

> > > > ### Author Response · Authors · 2023-08-21
> > > > **Response to Reviewer LyM3 [4/4]**
> > > >
> > > > ### Related Work
> > > >
> > > > > I don't see CLEVR or dSprites datasets mentioned in RW, although this benchmark builds on top of these.
> > > > >
> > > >
> > > > We have already cited CLEVR dataset in our related work section. The dSprites dataset was first used in the Beta-VAE paper [1] that we have also already discussed in our related work. Furthermore, the Spriteworld API (https://github.com/deepmind/spriteworld) that we used for making dSprites tasks is already referenced in our appendix.
> > > >
> > > > [1] *Irina Higgins, Loic Matthey, Arka Pal, Christopher Burgess, Xavier Glorot, Matthew Botvinick, Shakir Mohamed, and Alexander Lerchner*. beta-vae: Learning basic visual concepts with a constrained variational framework. In International conference on learning representations, 2017.
> > > >
> > > > > The relation to other similar datasets should be clearly stated.
> > > > >
> > > >
> > > > It would help us if the reviewer can point us to which datasets we have missed. We would be happy to add them.
> > > >
> > > > In fact, other reviewers have not found our related work to be a concern. They have noted that “related papers are adequately discussed” (A3fU), “difference from previous contribution is clearly discussed” (A3fU), and “related works are extensively cited and compared to.” (zB71)
> > > >
> > > > > I would give more focus to the existing benchmark datasets rather than to the methods as this works introduces a "novel" benchmark.
> > > > >
> > > >
> > > > We think there is nothing wrong in citing works that are related to ours but are not benchmarks. The relevance of a work should depend, not on its benchmark status, but on whether it asks a question related to systematic generalization.
> > > >
> > > > > There are many citations. Select only the most important ones.
> > > > >
> > > >
> > > > Given our work spans multiple sub-disciplines (OOD Generalization, Reasoning, Generative Modeling, Representation Learning, Video Prediction, World Modeling, and Reinforcement Learning), it is not surprising to have many related works. We think all our cited works are related and worth giving due credit.
> > > >
> > > > > How does your work relate to next-frame video prediction (e.g., KITTI benchmark), predictive coding, etc.
> > > > >
> > > >
> > > > We already discussed video prediction benchmarks (e.g., KITTI) in Appendix C. Different from our benchmark, existing video prediction benchmarks focus on the IID setting and don’t offer systematic out-of-distribution (OOD) testing splits for studying model generalization.
> > > >
> > > > ### Other Concerns
> > > >
> > > > > There is also no documentation on how to conduct your own experiments.
> > > > >
> > > >
> > > > As noted in the paper, we will open-source all code used to run our experiments and make datasets after acceptance.

---

> > > > > ### Comment · Reviewer_LyM3 · 2023-08-29
> > > > >
> > > > > Dear authors,
> > > > >
> > > > > Thank you for your long answer reflecting my concerns. With some parts of the answer I am satisfied and I am also sorry for overlooking some of the information in the paper. However, I have to admit that some of your answers and revisions in the paper are not for me satisfactory. Especially many of my points were to point you to the places where your paper is not clear and I would expect that you reflect your answers also in the revised paper..
> > > > > 1) I didn’t misunderstand your intention to provide broad introduction. I just think that in the way it is presented it is not helping the reader to understand the contributions of your work. Broad introduction can be seen as a strength only if it is related to the paper idea and helps understanding the paper contributions. It should be to the reader always clear why you mention any piece of information. I think rewriting and clarifying your introduction and contributions of the paper would help a lot. As for contributions, maybe adding front figure showing the benchmark would help (focusing on different tasks involved - see below). Also the saved space on cutting down a bit inttroduction could be used for e.g., describing metrics etc.
> > > > >
> > > > > 2) I am happy you explained some of the statements in the introduction, such as what “Systematic compositionality originates from the fields of philosophy of language and linguistics” or "which simplifies addressing compositional systematicity" mean. However, this was to provide you examples about unclear parts of the introduction and these explanations should be (at least partly) reflected in a revision of the paper, which I think is not yet the case.
> > > > >
> > > > > 3) I am happy you explained a bit more how your benchmark is built using CLEVR, dSprites and CLEVRTex datasets. I think you should highlight in the paper which tasks (the 4 benchmark rules) are in your benchmark. Either as a figure, table, or at least in the text. Since this is important for the benchmark, the description of those 4 categories of rules should be part of the main paper (especially when the evaluation is also shown towards these (see Fig.3). Whoever comes to the paper should clearly and quickly see what is the coverage of tasks of this benchamark (I guess front figure would be great) e.g. as a matrix of tasks vs. object code bases.
> > > > >
> > > > > 4) As for the metrics - thanks for the response, again, I mentioned it because it was not clearly stated in the text and I would find it way better to have this reflected in the paper. Right now, it is left on the reader to think what you exactly mean.
> > > > >
> > > > > 5) Baselines - thank you for clarification. I checked the appendix in a more detail and I think the description of the baselines there is clear. I understand that you cannot fit everything to the main paper so this part is ok, maybe try to highlight in the main paper at least names of baselines so it is easier to read. Sorry for overlooking Oracle description (I didn’t expect it to be part of the paragraph and not highlighted).
> > > > >
> > > > > 6) Alpha parameter - I see it now in the Section 2 after you pointed me to it, but I definitely didn’t remember it when getting to section 6 by myself.’ Again, take it as information for you on how to improve readability of the paper.
> > > > >
> > > > > Finally, making the codebase and benchmark available upon acceptance seems to me as the most weak point of the paper, especially in the track called “Datasets and benchmarks”. Accepting a paper upon a partial code seems more like accepting it based on “advertisement” rather than real work done. As a reviewer I am responsible to evaluate especially the quality of the benchmark and the codebase, which is not in this case possible.

---

> > > > > > ### Author Response · Authors · 2023-08-30
> > > > > >
> > > > > > > **General clarity of the writing**
> > > > > > >
> > > > > >
> > > > > > Thank you for your suggestion. We found it helpful and will improve the paper in the next revision based on your suggestions. However, we would like to emphasize that none of the other reviewers raised any concerns regarding the clarity of the writing. On the contrary, they all noted that the paper was "well-written and easy to read". Therefore, we respectfully disagree that the paper's clarity is unclear enough to warrant a score of 3.
> > > > > >
> > > > > > > **Add a figure in the main paper showing the benchmark and focusing on different tasks.**
> > > > > > >
> > > > > >
> > > > > > Thank you for the suggestion. In our new revision, we have added a figure summarizing all 12 tasks in the main paper (see Figure 3).
> > > > > >
> > > > > > > **Better describe LPIPS metric in the paper.**
> > > > > > >
> > > > > >
> > > > > > Thank you for the suggestion. We have now added a more detailed description of LPIPS in the main paper (see Lines 153-157).
> > > > > >
> > > > > > > **Make the descriptions of baselines and the alpha parameter easier to discover.**
> > > > > > >
> > > > > >
> > > > > > Thank you for these suggestions. In our new revision, we have made the following updates to address them:
> > > > > >
> > > > > > - Lines 174-176: We have now made a separate subsection for the Oracle baseline.
> > > > > > - Lines 184, and 187: We have highlighted the baselines SSM-VAE and SSM-Slot by italicizing them.
> > > > > > - Line 114: We renamed the relevant subsection to make it easier to discover the description of $\alpha$.
> > > > > >
> > > > > > > **Making the codebase and benchmark available upon acceptance seems to me as the most weak point of the paper. Only partial benchmark released. Evaluating the benchmark not possible.**
> > > > > > >
> > > > > >
> > > > > > **Regarding the Codebase.** We have now made the codebase available for anonymous review at the link [1]. The codebase contains all the code needed for running our experiments and creating our benchmark.
> > > > > >
> > > > > > **Regarding the Benchmark.** As mentioned earlier, we have already anonymously shared 100 instances of all splits for all our 12 tasks via the link [1]. Since the instances within a split are random samples generated from the same distribution, the shared preview is adequate for reviewing the benchmark. As mentioned earlier, we are committed to releasing the full benchmark within a week of acceptance. Importantly, this aligns with the recommendation on the Call for Papers page [2]: *"…We highly recommend making the dataset publicly available **immediately or before the start of the NeurIPS conference**.”*
> > > > > >
> > > > > > [1] https://github.com/systematic-visual-imagination/benchmark
> > > > > >
> > > > > > [2] https://nips.cc/Conferences/2023/CallForDatasetsBenchmarks

---

> > > > > > > ### Comment · Reviewer_LyM3 · 2023-08-30
> > > > > > >
> > > > > > > I would like to thank the authors for the changes in the manuscript, especially the figure summarizing the tasks, which I greatly appreciate. Similarly, I think that with the code, I can now better evaluate the benchmark's scope and quality. The benchmark (and number of rules) seems still quite simplistic as it focuses basically only on shape, color, and position switching, but after a quick check on the code (data_creation) in codebase, adding new rules should be rather easy. I just wonder why you don't cover all the combinations (e.g., apart of atomic rule for shape also atomic rule for color). This could give more insights on the compositional results.
> > > > > > >
> > > > > > > I think that it could be very interesting to enable the users to interactively create new rules (e.g. by adding a simple script via some interface) so they can easily create new benchmarking tasks. This could be very useful for wider usage by the community.
> > > > > > >
> > > > > > > Although I have still some small doubts about the work, I appreciate the great effort of authors to improve and explain their work and I adjust my rating score accordingly.

---

> > > > > > > > ### Author Response · Authors · 2023-08-30
> > > > > > > > **Thank You**
> > > > > > > >
> > > > > > > > Thank you, we are glad that our modifications and the figure have been helpful! We also thank you for the updated rating.
> > > > > > > >
> > > > > > > > > I just wonder why you don't cover all the combinations (e.g., apart from the atomic rule for shape).
> > > > > > > > >
> > > > > > > >
> > > > > > > > This is a great question. All combinations cannot be covered because the space of all possible rules is prohibitively large, amounting to about 65K causal graphs for dSprites alone. Therefore, we first built a set of 16 analysis rules, tested our baselines on these rules in the dSprites environment, selected 4 rules based on this analysis, and used these 4 selected rules to create the datasets for all 3 environments. In Appendix D.1.2, we provide the details about rule design and the considerations involved.
> > > > > > > >
> > > > > > > > > I think that it could be very interesting to enable the users to interactively create new rules.
> > > > > > > > >
> > > > > > > >
> > > > > > > > Thank you, this is an interesting aspect! We will consider such a possibility for future extensions of the benchmark.

---

### Author Response · Authors · 2023-08-21
**General Response**

We thank all the reviewers for their thoughtful reviews!

We are delighted that the reviewers found our benchmark to **address a fundamental issue** (A3fU) and present an **interesting problem** (LyM3, zB71). They recognized its **potential to inspire future research** (A3fU), commended it as a **useful evaluation tool** (zB71), and noted that it **can make a clear contribution** to the research field (ikkV). They found our **baseline models well-designed** (A3fU) and appreciated our providing a performance upper-bound (zB71). They also found the paper to be **well-written** (A3fU, ikkV) and **easy to read and follow** (A3fU, zB71) with our documentation **clearly organized** and **good for reproducibility** (ikkV, A3fU). Our related work also received positive comments such as **adequately discussed** (A3fU) and **extensively cited** (zB71).

### List of Revisions

We provide a list of revisions made to the manuscript based on reviewer comments and suggestions.

1. Section 2.1: Clarified the acronym OOD.
2. Table 4: Added a table reporting accuracy of predicting the ground-truth factors from CLEVRTex images.
3. Section 7.1: Added a section on limitations
4. Section 6.3: Modified this section to better highlight pointers for improving the baselines.
5. Appendix C: Added a discussion of works that pursue large-scale pretraining with visual prompting.
6. Section D.2: Added an algorithm describing the process to construct the core combinations.
7. Section D.1.2: Added clarification about the analysis experiments.
8. Figure 17: Expanded the caption to clarify the underlying rules for the 16 analysis tasks.

We now respond to each reviewer individually.

---

### Decision · Program_Chairs · 2023-09-22

**Decision:**

Accept (Poster)

**Comment:**

This paper introduces a new benchmark for evaluating systematic compositionality in visual generative models in the form of a one-step image-to-image generation task as well as a thorough empirical analysis of baseline models on this dataset. The proposed dataset is simple to analyze and enables evaluation of this ability in isolation without it getting muddled with other spurious factors of variation that might exist in real-world images, which I see as a strength. But as pointed out by some reviewers, there does exist a gap in visual realism, which means that it's important for the proposed dataset to be used as a litmus test while developing generative models and not for it to become an end benchmark to optimize in isolation. Overall, this is a solid contribution, and I appreciate the authors for engaging with the reviewers and being proactive about incorporating suggestions. I recommend the paper for acceptance.